# Breaking Independence: Learning View Correlation for Variational Incomplete Multi-View Clustering

## Abstract

Incomplete multi-view clustering (IMVC) aims to uncover shared cluster structures from data with partially observed views. Although recent imputation-free methods based on variational inference demonstrate robustness to missing views, they commonly rely on a conditional independence assumption across views, which fails to capture the inherently structured and potentially correlated nature of multi-view data. In this paper, we propose a variational framework that explicitly breaks this assumption by introducing a learnable cross-view correlation structure. Specifically, we explicitly model and learn correlations between views by utilizing the covariance structure of posterior estimation errors. To facilitate robust and efficient learning, the correlation matrix is parameterized through a normalized Cholesky decomposition, ensuring positive definiteness and enabling the entire model to be trained jointly through a unified variational objective. Extensive experiments on multiple IMVC benchmarks demonstrate that our method consistently outperforms state-of-the-art approaches across a wide range of missing-view settings. These results highlight the effectiveness of adaptive correlation modeling in variational IMVC, demonstrating the need to break the independence assumption.

## 1 Introduction

Multi-view clustering (MVC) aims to partition data objects that are characterized by multiple distinct feature sets or views into their natural groups. However, in real-world scenarios, missing views are prevalent due to factors such as sensor failures, data corruption, or privacy concerns, giving rise to the problem of incomplete multi-view clustering (IMVC). Among existing approaches, imputation-based IMVC methods have gained significant attention by recovering missing data or latent features (Lin et al., 2023; Wang et al., 2022b; Zhang et al., 2025b; Wen et al., 2024; Gao & Pu, 2025) before clustering. Nevertheless, the main challenge in imputation-based methods lies in the accurate recovery of missing data without ground-truth supervision, especially when the proportion of missing views is large.

To avoid this challenge, imputation-free IMVC methods have emerged to directly learn from the available data. These include graph-based methods (Wen et al., 2023; 2020b), cross-view mapping methods (Xu et al., 2022; Zhao et al., 2025), and distribution alignment methods (Xu et al., 2023). Among them, DVIMC (Xu et al., 2024) pioneered the application of Product of Experts (PoE) for IMVC, standing out for its effectiveness in learning robust shared latent representations in the missing views settings. However, these methods treat each view-specific posterior as independent, an assumption that oversimplifies the rich dependencies that naturally exist among views. Effective fusion requires that views compensate for missing or unreliable information in one another, but also that the model account for correlations across views, since highly correlated views are prone to making similar errors. A recent supervised approach, CoDE (Mancisidor et al., 2025), revisits the classic consensus-of-experts formulation (Winkler, 1981) and shows that explicitly modeling inter-expert correlations through their estimation errors is highly beneficial and provides a principled Bayesian approach. This perspective enables the model to capture cross-expert dependencies without explicitly parameterizing the relationships between the experts themselves, which is non-trivial as the joint posterior being approximated is a distribution of a latent variable. However, CoDE relies on a fixed scalar correlation obtained through a supervised grid search, rendering it impractical for IMVC,

where labels are unavailable and views are incomplete. Moreover, their fixed correlation value fails to capture the diverse, pairwise dependencies across views, thereby compromising their performance.

To tackle the aforementioned problems, we propose an **A**daptive **CO**rrelation-aware **V**ariational **A**ggregation framework for IMVC (ACOVA), which models inter-view dependencies via learning the estimation error covariance under missing scenarios. Specifically, we go beyond the strong independence assumption and extend the existing variational framework by Xu et al. (2024) to a more realistic setting, in which the view-specific estimation errors are assumed to be correlated, allowing better capturing of the inherent dependencies between views. Furthermore, instead of relying on a supervised grid search to tune a global scalar correlation, which is infeasible in IMVC, we parameterize the correlation matrix via a normalized Cholesky decomposition, allowing it to be learned jointly with the model parameters through the overall training objective.

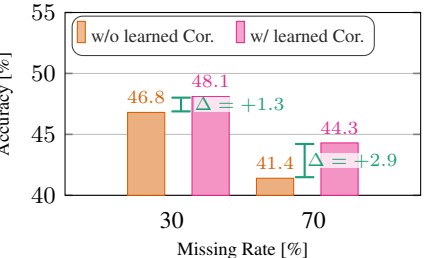

**Figure 1:** Benefit of modeling inter-view dependencies for different missing rates on Scene15.

In summary, our major contributions are as follows:
(1) We propose ACOVA, a simple yet effective correlation-aware variational framework for IMVC that explicitly models inter-view correlations through the estimation error covariance, going beyond the independence assumption in posterior aggregation that limits prior IMVC methods. (2) We develop an adaptive cross-view correlation learning mechanism to jointly learn the correlation matrix and model parameters, leading to improved performance (see Figure 1). (3) We provide an extensive empirical evaluation, with our method achieving leading performance on multiple IMVC benchmarks.

## 2 RELATED WORK

**Incomplete Multi-View Clustering.** The development of deep neural networks has led to significant progress in deep IMVC (Wen et al., 2020a; Li et al., 2022; Wang et al., 2022a; Li et al., 2021), resulting in two main types of methods: imputation-based and imputation-free approaches. Imputation-based methods aim to complete missing data through various strategies before performing clustering on the reconstructed multi-view dataset. Lin et al. (2023; 2021) utilize contrastive prediction combined with entropy-based optimization to recover missing views, while other methods adopt prototype-based alignment and imputation (Yuan et al., 2025) or apply diffusion models to generate missing samples (Wen et al., 2024; Zhang et al., 2025b;a). However, these methods have the underlying issue that inaccurate or biased imputations will negatively affect the subsequent representation learning and clustering step. More recently, imputation-free methods (Jin et al., 2023; Wen et al., 2023) have therefore started to emerge, which avoid explicit missing data recovery and instead learn from available view-specific representations directly for downstream clustering. For example, CDIMC (Wen et al., 2020b) leverages graph embeddings and a self-paced learning strategy to enhance shared representations, while DIMVC (Xu et al., 2022) employs nonlinear mappings across views to exploit complementary information. APADC (Xu et al., 2023) emphasizes aligning the distributions of available data during feature learning, and DVIMC (Xu et al., 2024) incorporates variational autoencoders with coherence constraints to improve learning of a common representation.

**Variational approaches for Incomplete Multi-View Clustering.** Variational Autoencoders (VAEs) (Kingma & Welling, 2014) provide an effective framework for learning latent distributions by incorporating diverse prior distributions and generative assumptions, and have been extensively applied to clustering (Jiang et al., 2017; Yang et al., 2019; Falck et al., 2021; Lim et al., 2020). Within multi-view clustering, numerous studies (Cui et al., 2023; Huang et al., 2023; Yin et al., 2020) have extended VAEs with principled approaches to aggregate information across views and derive joint representations. For instance, Multi-VAE (Xu et al., 2021) disentangles shared and view-specific features to improve clustering performance. However, most existing approaches are not designed for incomplete multi-view scenarios, limiting their practical applications. To address this challenge, a recent method DVIMC (Xu et al., 2024) employs a product-of-experts strategy but assumes independence across views, limiting its ability to capture inter-view relationships. MVP (Gao & Pu, 2025) utilizes cyclic permutations within VAEs but relies on handcrafted priors and

complex partitioning schemes that limit scalability. In contrast, our method adaptively learns a variational posterior with a cross-view correlation structure, enabling more expressive modeling of inter-view dependencies without explicit permutations.

# 3 PRELIMINARIES ON VARIATIONAL INCOMPLETE MULTI-VIEW CLUSTERING

In the following, we briefly review the preliminaries of variational IMVC; additional background is provided in Appendix A.1.

**Notations** Let $\{\mathcal{X}\}$ be an incomplete multi-view dataset, where we observe $\{\mathbf{x}_v\}_{v\in\mathcal{V}}$, and $\mathcal{V}\subseteq[V]$ denotes the set of available views. Here, $[V]=\{1,2,\ldots,V\}$ represents the complete view set. Each $\mathbf{x}_v\in\mathbb{R}^{D_v}$ lies in a view-specific space. For brevity, we denote the observed view set of a generic instance by $\mathcal{V}$, with $\{\mathbf{x}_v\}_{v\in\mathcal{V}}$ representing the available multi-view inputs. In the missing-view setting, $\mathbf{x}_v$ denotes the data available for view $v$. The goal of IMVC is to partition the $N$ instances into $C$ clusters.

**Generative Process** We assume that the multi-view data is generated from a random process involving latent variables $\mathbf{z}\in\mathbb{R}^D$ and discrete latent variables $\mathbf{c}\in\{1,\ldots,C\}$ that denote the common representation and cluster assignment, respectively. In the incomplete setting, assuming that the observed data are conditionally independent given the shared latent representation $\mathbf{z}$, the joint probability of the observed data and latent variables can be formulated as:

$$p(\{\mathbf{x}_v\}_{v=1}^V, \mathbf{z}, \mathbf{c}) = p(\mathbf{c})\,p(\mathbf{z}\mid\mathbf{c})\,p(\{\mathbf{x}_v\}_{v=1}^V\mid\mathbf{z}) = p(\mathbf{c})\,p(\mathbf{z}\mid\mathbf{c})\prod_{v=1}^V p(\mathbf{x}_v\mid\mathbf{z}). \tag{1}$$

**Inference Objective** Our goal is to infer the joint posterior distribution $p(\mathbf{z}, \mathbf{c}|\{\mathbf{x}_v\}_{v\in\mathcal{V}})$, which can be expressed through Bayes' theorem as:

$$p(\mathbf{z}, \mathbf{c}|\{\mathbf{x}_v\}_{v\in\mathcal{V}}) = \frac{p(\{\mathbf{x}_v\}_{v\in\mathcal{V}}|\mathbf{z})p(\mathbf{z}, \mathbf{c})}{\int_{\mathbf{z}}\sum_{\mathbf{c}=1}^C p(\{\mathbf{x}_v\}_{v\in\mathcal{V}}|\mathbf{z}, \mathbf{c})p(\mathbf{z}, \mathbf{c})d\mathbf{z}}. \tag{2}$$

Due to the intractability of this posterior, variational inference introduces an approximate posterior $q_\phi(\mathbf{z}, \mathbf{c}|\{\mathbf{x}_v\}_{v\in\mathcal{V}})$ by minimizing the Kullback-Leibler (KL) divergence:

$$D_{\mathrm{KL}}(q_\phi(\mathbf{z}, \mathbf{c}|\{\mathbf{x}_v\}_{v\in\mathcal{V}})||p(\mathbf{z}, \mathbf{c}|\{\mathbf{x}_v\}_{v\in\mathcal{V}})). \tag{3}$$

It is common to assume $\mathbf{z}$ and $\mathbf{c}$ are independent given $\{\mathbf{x}_v\}_{v\in\mathcal{V}}$, which allows to factorize the joint distribution as $q_\phi(\mathbf{z}, \mathbf{c}|\{\mathbf{x}_v\}_{v\in\mathcal{V}}) = q_\phi(\mathbf{z}|\{\mathbf{x}_v\}_{v\in\mathcal{V}})q_\phi(\mathbf{c}|\{\mathbf{x}_v\}_{v\in\mathcal{V}})$.

**Joint Posterior Aggregation** With the VAE paradigm, we approximate the posterior $q_\phi(\mathbf{z}|\mathbf{x}_v) = \mathcal{N}(\mathbf{z}|\boldsymbol{\mu}_v, \boldsymbol{\Sigma}_v)$ of each view using a Gaussian distribution[1], where $\boldsymbol{\mu}_v$ and $\boldsymbol{\sigma}_v^2$ are the mean and variance parameters inferred by the $v$-th view-specific encoders. Based on this, the approximate posterior $q_\phi(\mathbf{z}|\{\mathbf{x}_v\}_{v\in\mathcal{V}})$ is typically obtained through an aggregation mechanism $\mathcal{A}$ that combines view-specific posteriors $q_\phi(\mathbf{z}|\mathbf{x}^{(v)})$:

$$\mathcal{A}: \{\boldsymbol{\mu}_v, \boldsymbol{\sigma}_v^2\}^{|\mathcal{V}|} \to \{\tilde{\boldsymbol{\mu}}, \tilde{\boldsymbol{\sigma}}^2\}. \tag{4}$$

Following this paradigm, recent multi-view clustering methods primarily differ in how they derive the joint posterior variational distribution $\mathcal{N}(\tilde{\boldsymbol{\mu}}, \tilde{\boldsymbol{\sigma}}^2)$. DMVCVAE (Yin et al., 2020) emphasizes the varying importance of different views and proposes a weighted combination approach in the complete multi-view setting. A more relevant approach, DVIMC (Xu et al., 2024), tackles the incomplete multi-view setting using the Product of Experts (PoE) strategy, which treats each view as an independent expert and combines them through precision-weighted averaging. However, both methods rely on a

---

[1]This research assumes a diagonal covariance matrix for all variational distributions, i.e. $\boldsymbol{\Sigma} = \boldsymbol{\sigma}^2\mathbf{I}$.

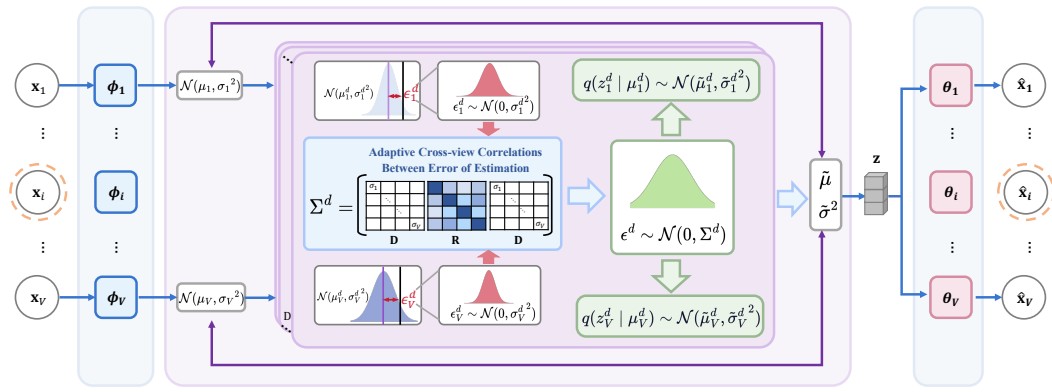

**Figure 2:** Overview of our proposed **ACOVA** framework. Given multi-view inputs $\{\mathbf{x}_v\}_{v=1}^{V}$, view-specific encoders $\{\phi_v\}$ produce posterior distribution parameters $\{\boldsymbol{\mu}_v, \boldsymbol{\sigma}_v^2\}$ to characterize view-wise uncertainty. For each dimension $d \in D$, the estimation errors $\epsilon_v^d$ between the view-specific posteriors and the true latent variable are modeled with learnable cross-view correlations, captured by the correlation matrix $\mathbf{R}$. This structure, together with view-specific variances $\mathbf{D}$, forms a structured covariance matrix $\boldsymbol{\Sigma}^d = \mathbf{DRD}$, which governs the aggregation of posteriors into a consensus distribution with parameters values $\{\tilde{\mu}_v^d, \tilde{\sigma}_v^{d\,2}\}$. A latent representation $\mathbf{z} \in \mathbb{R}^D$ is sampled from this aggregated posterior and decoded via view-specific decoders $\{\theta_v\}$ for reconstruction. Note that conditioning on $\boldsymbol{\mu}_v$ or on $\mathbf{x}_v$ is equivalent in this context.

strong independence assumption across views, which is often overly simplistic in real-world scenarios where views are semantically correlated. This limits the capacity of the model to capture cross-view interactions and express the true joint posterior. Besides, both approaches have drawbacks in the aggregation of the joint posterior. DMVCVAE (Yin et al., 2020) could be seen as a partial realization of the Mixture of Experts (MoE), where only the within-view variances are considered, omitting the variability arising from disagreements between views, leading to underestimated overall uncertainty. While in DVIMC (Xu et al., 2024), PoE is known to underestimate the variance of the consensus distribution due to its strong independence assumption. Let $\boldsymbol{\xi}_v = 1/\boldsymbol{\sigma}_v^2$ denote the precision of view $v$, the aggregated precision under PoE could then be denoted as $\tilde{\boldsymbol{\xi}}_{\text{PoE}} = \sum_v \boldsymbol{\xi}_v$. As a result, the aggregated variance becomes $\tilde{\boldsymbol{\sigma}}^2 = 1/\tilde{\boldsymbol{\xi}}_{\text{PoE}}$, which is always smaller than any individual $\boldsymbol{\sigma}_v^2$. This issue becomes especially concerning in the incomplete multi-view scenario, where limited observations naturally lead to higher uncertainty, requiring more cautious and calibrated posterior variance estimation.

## 4 ADAPTIVE CORRELATION-AWARE VARIATIONAL AGGREGATION

As discussed previously, IMVC requires posterior inference over samples with arbitrarily missing views. A central challenge lies in how to aggregate view-specific posterior distributions to form a shared latent representation, particularly when the common independence assumption is overly simplistic and leads to underestimation of the posterior variance. Inspired by (Mancisidor et al., 2025; Winkler, 1981), we model the inter-view dependence through the error of estimation associated with each view's posterior distribution. However, directly applying these methods to IMVC scenarios poses significant limitations. On the one hand, they barely learns the covariance matrix of the error of estimation but cross-validate the implicit correlation, which is infeasible in unsupervised clustering settings where ground-truth labels are unavailable. On the other hand, based on grid-searching, they assume a fixed correlation, which restricts the model's ability to capture diverse and pairwise specific dependencies across views. To address these problems, as shown in Figure 2, we propose a novel approach in which the correlation structure is learned jointly with the model parameters, enabling modeling of adaptive inter-view dependencies under missing-view scenarios without supervision information.

### 4.1 POSTERIOR DERIVATION WITH ADAPTIVE CROSS-VIEW CORRELATIONS

Given the mean parameters $\{\boldsymbol{\mu}_v\}_{v \in \mathcal{V}}$ of the view-specific variational distributions, we define $\boldsymbol{\mu}^d = [\mu_1^d, \mu_2^d, \ldots, \mu_V^d]^\top \in \mathbb{R}^{V \times 1}$ as the vector of view-specific estimates for the $d$-th dimension of the

latent variable $\mathbf{z}$, where $\mu_v^d$ represents the estimate of view $v$ for dimension $d$. The complete estimate vector is then constructed as $\boldsymbol{\mu} = [\boldsymbol{\mu}^{1\top}, \boldsymbol{\mu}^{2\top}, \ldots, \boldsymbol{\mu}^{D\top}]^\top \in \mathbb{R}^{VD \times 1}$ by concatenating all dimension-wise estimates. Given variance parameters $\{\sigma_v^2\}_{v \in \mathcal{V}}$, $\boldsymbol{\sigma}^2$ is defined similarly. To model the discrepancy between the estimate $\boldsymbol{\mu}$ and the true, but unknown, latent variable $\mathbf{z}$, Definition 4.1 introduces the error of estimation (Winkler, 1981; Mancisidor et al., 2025).

**Definition 4.1** (Error of Estimation). *Given the estimate $\boldsymbol{\mu} = [\boldsymbol{\mu}^{1\top}, \boldsymbol{\mu}^{2\top}, \ldots, \boldsymbol{\mu}^{D\top}]^\top$ by the view-specific distributions $q_v(\mathbf{z}|\mathbf{x}_v) = \mathcal{N}(\boldsymbol{\mu}_v, \boldsymbol{\sigma}_v^2)$, the error of estimation of the latent variable $\mathbf{z}$ can be formulated as:*

$$\boldsymbol{\epsilon} = \boldsymbol{\mu} - \mathbb{1}\mathbf{z}, \tag{5}$$

*where $\mathbb{1} \in \mathbb{R}^{VD \times D}$ is a block diagonal design matrix with structure $\mathbb{1} = blkdiag(\mathbf{1}_V, \mathbf{1}_V, \ldots, \mathbf{1}_V)$ containing $D$ blocks, each $\mathbf{1}_V \in \mathbb{R}^{V \times 1}$ being a column vector of ones, $\boldsymbol{\epsilon} \sim \mathcal{N}(\mathbf{0}, \boldsymbol{\Sigma})$, and $\boldsymbol{\Sigma} = \mathrm{diag}[\boldsymbol{\Sigma}^1, \boldsymbol{\Sigma}^2, \cdots, \boldsymbol{\Sigma}^D] \in \mathbb{R}^{VD \times VD}$ consists of the error of estimation covariance matrices $\boldsymbol{\Sigma}^d \in \mathbb{R}^{V \times V}$ for each dimension $d$. Here, $\boldsymbol{\Sigma} = \mathrm{diag}[\boldsymbol{\Sigma}^1, \boldsymbol{\Sigma}^2, \cdots, \boldsymbol{\Sigma}^D]$ consists of error of estimation covariance matrices for each dimension $\boldsymbol{\Sigma}^d$.*

Based on the above definition and as shown in Figure 2, each of the available view-specific distributions provides an assessment $\boldsymbol{\mu}^d$ of the joint latent variable $\mathbf{z}^d$. Therefore, the covariance matrix $\boldsymbol{\Sigma}^d$ reflects the interdependency between the error of estimation of the view-specific distributions. This dependency should not be ignored in previous aggregation methods, since all distributions are exposed to the same underlying object. It is worth mentioning that Mancisidor et al. (2025) learns the same dependency correlations across views, but relies on supervised cross-validation to grid-search the implicit correlation between the assessments of the view-specific distributions. However, this supervised approach is incompatible with unsupervised clustering tasks, where ground-truth labels are unavailable for tuning correlation parameters. Therefore, in this paper, we propose to adaptively learn the cross-view correlation structure through a novel differentiable parameterization:

**Definition 4.2** (Adaptive Cross-view Correlation Learning). *Let $\mathbf{L}$ be a learnable lower-triangular matrix with positive diagonal elements ensured by exponentiation. We use the Cholesky decomposition of the covariance matrix $\boldsymbol{\Sigma}^d$ to guarantee the positive definiteness constraint and bounded entries of $\mathbf{R} = [\rho_{ij}]_{i,j=1}^V \succ 0$, a positive definite correlation matrix with $\rho_{ii} = 1$ and $\rho_{ij} \in [-1, 1]$ for $i \neq j$, as follows:*

$$\mathbf{R}_{ij} = \frac{(\mathbf{L}\mathbf{L}^\top)_{ij}}{\sqrt{(\mathbf{L}\mathbf{L}^\top)_{ii}(\mathbf{L}\mathbf{L}^\top)_{jj}}}. \tag{6}$$

*We use the following decomposition of error covariance $\boldsymbol{\Sigma}^d$ for the d-th dimension*

$$\boldsymbol{\Sigma}^d = \begin{bmatrix} \sigma_1^2 & \rho_{12}\sigma_1\sigma_2 & \cdots & \rho_{1V}\sigma_1\sigma_V \\ \rho_{21}\sigma_2\sigma_1 & \sigma_2^2 & \cdots & \rho_{2V}\sigma_2\sigma_V \\ \vdots & \vdots & \ddots & \vdots \\ \rho_{V1}\sigma_V\sigma_1 & \rho_{V2}\sigma_V\sigma_2 & \cdots & \sigma_V^2 \end{bmatrix} = \mathbf{D}\mathbf{R}\mathbf{D}, \tag{7}$$

*where $\mathbf{D} = diag(\sigma_1, \ldots, \sigma_V)$ contains the view-specific standard deviations, to learn pairwise dependencies between view-specific variational distributions.*

Accordingly, by decomposing the error covariance $\boldsymbol{\Sigma}^d = \mathbf{D}\mathbf{R}\mathbf{D}$, we provide a general framework that unifies existing approaches through the separation of variance $\mathbf{D}$ and correlation matrix $\mathbf{R}$. When $\mathbf{R} = \mathbf{I}$, our method degenerates to DVIMC (Xu et al., 2024) with independence assumption; when $\mathbf{R} = \rho\mathbf{I}$ for a fixed scalar $\rho$, it degenerates to CoDE (Mancisidor et al., 2025) with a uniform correlation. Our approach generalizes beyond these limitations by modeling a full correlation matrix $\mathbf{R}$ that can adaptively model heterogeneous dependencies between views through a learnable Cholesky decomposition parameterization, enabling end-to-end optimization without manual tuning.

To understand $\mathbf{R}$'s capacity to deviate from independence-based aggregation, we provide a theoretical deviation bound from the identity matrix:

**Theorem 4.1** (Frobenius Identity Deviation Bound). *Given $\mathbf{R} = [\rho_{ij}]_{i,j=1}^V \succ 0$ where $\rho_{ii} = 1$ and $\rho_{ij} \in [-1, 1]$ for $i \neq j$, the Frobenius deviation from identity matrix $\mathbf{I}$ is upper-bounded as*

$$\|\mathbf{R} - \mathbf{I}\|_F \leq \sqrt{V(V-1)}. \tag{8}$$

The proof of the bound can be found in Appendix A.1. The bound formally characterizes the maximal Frobenius deviation of a valid correlation matrix from the identity matrix, offering a theoretical guarantee on the extent of structural variability induced by cross-view dependency modeling. We further provide theoretic results demonstrating the identifiability of the learned correlations in Appendix A.3.

Based on the above analysis, denoting a binary mask matrix as $\mathbf{M} \in \{0, 1\}$ to indicate the availability of view-specific observations, the joint posterior distribution for incomplete multi-view data can then be formulated as (see derivation in Appendix):

$$q(\mathbf{z} \mid \{\mathbf{x}_v\}_{v \in \mathcal{V}}) \sim \mathcal{N}(\tilde{\boldsymbol{\mu}}, \tilde{\boldsymbol{\sigma}}^2) = \mathcal{N}(\mathbf{A}_{\mathbf{M}}^{-1}\mathbf{B}_{\mathbf{M}}, \mathbf{A}_{\mathbf{M}}^{-1}) \tag{9}$$

where the aggregation matrices are computed as:

$$\mathbf{A}_{\mathbf{M}} = \mathbb{1}^\top (\boldsymbol{\Sigma}^{-1} \odot \mathbf{M}\mathbf{M}^\top) \mathbb{1} \tag{10}$$

$$\mathbf{B}_{\mathbf{M}} = \mathbb{1}^\top (\boldsymbol{\Sigma}^{-1} \odot \mathbf{M}\mathbf{M}^\top)(\boldsymbol{\mu} \odot \mathbf{M}) \tag{11}$$

Here, the posterior aggregation is controlled by two matrices derived from the correlation structure and mask, where $\mathbf{A}_{\mathbf{M}}$ represents the precision matrix of the aggregated posterior, and $\mathbf{B}_{\mathbf{M}}$ contains the precision-weighted mean estimates. The Hadamard product $\odot$ with the mask ensures that only available observations contribute to the aggregation process.

## 4.2 ELBO FOR INCOMPLETE MULTI-VIEW CLUSTERING

The cluster assignment posterior $q_\phi(\mathbf{c} | \{\mathbf{x}_v\}_{v \in \mathcal{V}})$ can be approximated though the VaDE (Jiang et al., 2017) trick by $q_\phi(\mathbf{z} | \{\mathbf{x}_v\}_{v \in \mathcal{V}})$ via Bayes' rule:

$$q_\phi(\mathbf{c} | \{\mathbf{x}_v\}_{v \in \mathcal{V}}) = q(\mathbf{c} \mid \mathbf{z}) \equiv \frac{p(\mathbf{c})\, p(\mathbf{z} \mid \mathbf{c})}{\sum_{\mathbf{c}} p(\mathbf{c})\, p(\mathbf{z} \mid \mathbf{c})}, \tag{12}$$

where $p(\mathbf{c}) = \mathrm{Cat}(\boldsymbol{\tau})$ and $p(\mathbf{z} \mid \mathbf{c}) \sim \mathcal{N}(\mathbf{z} \mid \boldsymbol{\mu}_c, \boldsymbol{\Sigma}_c)$ being a Gaussian mixture prior. Here, $\mathrm{Cat}(\boldsymbol{\tau})$ refers to the categorical distribution with $\boldsymbol{\tau_c}$ ($\sum \boldsymbol{\tau_c} = 1$) being the prior probability for cluster $c$. Then the evidence lower bound (ELBO) derived from Eq. 3 can be expressed as[2]:

$$\mathcal{L}_{\mathrm{ELBO}}(\{\mathbf{x}_v\}_{v \in \mathcal{V}}) = \mathbb{E}_{q_\phi(\mathbf{z}|\{\mathbf{x}_v\}_{v \in \mathcal{V}})} \left[ \sum_{v \in \mathcal{V}} \log p(\mathbf{x}_v \mid \mathbf{z}) \right]$$

$$- \mathbb{E}_{q_\phi(\mathbf{c}|\{\mathbf{x}_v\}_{v \in \mathcal{V}})} [D_{\mathrm{KL}} (q_\phi(\mathbf{z} \mid \{\mathbf{x}_v\}_{v \in \mathcal{V}}) \,\|\, p(\mathbf{z} \mid \mathbf{c}))] - D_{\mathrm{KL}} (q_\phi(\mathbf{c} \mid \{\mathbf{x}_v\}_{v \in \mathcal{V}}) \,\|\, p(\mathbf{c})). \tag{13}$$

## 4.3 TRAINING OBJECTIVE

Following (Xu et al., 2024), we include a distribution alignment loss to encourage consistency between the aggregated posterior and the individual view-specific posteriors:

$$\mathcal{L}_{\mathrm{align}} = -\frac{1}{|\mathcal{V}|} \sum_{v \in \mathcal{V}} D_{\mathrm{KL}} \left( q_\phi \left( \mathbf{z} \mid \{\mathbf{x}_v\}_{v=1}^{V} \right) \big\| q_{\phi_v} (\mathbf{z} \mid \mathbf{x}_v) \right). \tag{14}$$

Therefore, with balancing hyperparameter $\alpha$, the total training objective is:

$$\mathcal{L}_{\mathrm{total}} = \mathcal{L}_{\mathrm{ELBO}} + \alpha \mathcal{L}_{\mathrm{align}}. \tag{15}$$

**Optimization of correlation matrix R.** In Eq. 15, both of the KL divergences terms $D_{\mathrm{KL}} (q_\phi(\mathbf{z} \mid \{\mathbf{x}_v\}_{v \in \mathcal{V}}) \,\|\, p(\mathbf{z} \mid \mathbf{c}))$ and $\sum_{v \in \mathcal{V}} D_{\mathrm{KL}} \left( q_\phi \left( \mathbf{z} \mid \{\mathbf{x}_v\}_{v=1}^{V} \right) \big\| q_{\phi_v} (\mathbf{z} \mid \mathbf{x}_v) \right)$ can be seen as functions of $\mathbf{R}$. Assuming Gaussian distributions, their combined yields:

---

[2]The full derivation is included in the Appendix.

$$
-\frac{1}{2}\Bigg\{\underbrace{\mathrm{tr}\left(\boldsymbol{\Sigma}_c^{-1}\mathbf{A}_\mathbf{M}^{-1}\right) + (\boldsymbol{\mu}_c - \mathbf{A}_\mathbf{M}^{-1}\mathbf{B}_\mathbf{M})^\top\boldsymbol{\Sigma}_c^{-1}(\boldsymbol{\mu}_c - \mathbf{A}_\mathbf{M}^{-1}\mathbf{B}_\mathbf{M}) + \ln|\boldsymbol{\Sigma}_c| - \ln|\mathbf{A}_\mathbf{M}^{-1}|}_{\text{Cross-view Correlation Learning}}
$$

$$
+ \alpha\underbrace{\sum_{v\in\mathcal{V}}\left[\mathrm{tr}\left(\boldsymbol{\Sigma}_v^{-1}\mathbf{A}_\mathbf{M}^{-1}\right) + (\boldsymbol{\mu}_v - \mathbf{A}_\mathbf{M}^{-1}\mathbf{B}_\mathbf{M})^\top\boldsymbol{\Sigma}_v^{-1}(\boldsymbol{\mu}_v - \mathbf{A}_\mathbf{M}^{-1}\mathbf{B}_\mathbf{M}) + \ln|\boldsymbol{\Sigma}_v| - \ln|\mathbf{A}_\mathbf{M}^{-1}|\right]}_{\text{View-specific Consistency Alignment}}\Bigg\},
$$

$$(16)$$

which is the term being optimized in an end-to-end manner to learn the correlation matrix $\mathbf{R}$.

## 5 EXPERIMENT

### 5.1 EXPERIMENTAL SETUP

**Datasets.** The experiments are carried out on the following six standard IMVC datasets. **Handwritten** (LeCun et al., 1989) contains 2,000 digit samples ranging from '0' to '9', where each sample is described using six types of features extracted from handwritten numerals. **Caltech5V** (Fei-Fei et al., 2004) consists of 1,400 images across 7 object categories, with each image represented by five types of visual features. **Scene15** (Fei-Fei & Perona, 2005) includes 4,485 images covering 15 scene categories. Each image is represented using three different visual features. **Fashion-MV** (Xiao et al., 2017) is constructed from the Fashion-MNIST dataset, containing 10,000 images from 10 clothing categories. Each sample is represented from three different views. **NoisyMNIST** (Wang et al., 2015) is a large-scale dataset that consists of 2 views, 10 classes and 70,000 samples. **CUB Image-Caption** (Zhang et al., 2019) is a multi-modal dataset consisting of 600 images with corresponding text descriptions of 10 bird classes.

**Incomplete multi-view data processing.** Following previous works (Lin et al., 2023; Wen et al., 2020b; Xu et al., 2024), we construct the incomplete multi-view datasets by randomly selecting $p\%$ ($p = \{10, 30, 50, 70\}$) of the samples and randomly deleting views, under the constraint that every sample retains at least one available view. To ensure fair comparisons, results are averaged over five runs, with the same five masks being used across methods.

**Baselines.** In our experiments, we compare our ACOVA with eight recent state-of-the-art methods, namely **BSV** (Zhao et al., 2016), **CONCAT** (Zhao et al., 2016), **DCP** (Lin et al., 2023), **DSIMVC** (Tang & Liu, 2022), **MVP** (Gao & Pu, 2025), **CPSPAN** (Jin et al., 2023), **DIMVC** (Xu et al., 2022), and **PMIMC** (Yuan et al., 2025).

**Evaluation Measures.** We evaluate clustering performance using four widely adopted metrics for IMVC (Amigó et al., 2009): clustering accuracy (ACC), normalized mutual information (NMI), adjusted Rand index (ARI), and purity (PUR). Higher values indicate better clustering performance for all four metrics.

Additional experimental and implementation details can be found in Appendix B.

### 5.2 COMPARISON WITH STATE-OF-THE-ART METHODS

As shown in Table 1, we compare ACOVA with eight state-of-the-art IMVC methods across six real-world datasets. It can be observed that: (1) Compared to the rest of the methods, ACOVA consistently ranks first or second across all datasets and missing rate settings, validating the effectiveness of our adaptive correlation learning strategy. (2) Compared to existing variational IMVC methods, such as DVIMC, which builds upon a strong independence assumption, and MVP, which leverages cyclic permutation to impute latent variables, ACOVA achieves superior performance in most cases demonstrating the effectiveness of our cross-view correlation modeling and learning mechanism.

### 5.3 ABLATION STUDY

**Effectiveness of Learning the Correlation.** To investigate the effectiveness of learning cross-view correlations, we compare our method (Learn $\mathbf{R}$ (*w/* corr.)) to an ablated version (*w/o* corr.), where

**Table 1:** Clustering performance under various missing-view rates on all datasets. Best results and second best are highlighted in **bold** and underlined, respectively. Note that all results are averaged over five runs.

| | Missing Rate | 10% | | | | 30% | | | | 50% | | | | 70% | | | |
|---|---|---|---|---|---|---|---|---|---|---|---|---|---|---|---|---|---|
| | Metrics | ACC↑ | NMI↑ | ARI↑ | PUR↑ | ACC↑ | NMI↑ | ARI↑ | PUR↑ | ACC↑ | NMI↑ | ARI↑ | PUR↑ | ACC↑ | NMI↑ | ARI↑ | PUR↑ |
| **Scene15** | BSV | 0.3595 | 0.3577 | 0.1784 | 0.4002 | 0.3305 | 0.3206 | 0.1253 | 0.3630 | 0.2944 | 0.2905 | 0.0796 | 0.3321 | 0.2593 | 0.2568 | 0.0472 | 0.2903 |
| | Concat | 0.3703 | 0.3820 | 0.2037 | 0.4263 | 0.3360 | 0.3318 | 0.1540 | 0.3825 | 0.2943 | 0.2995 | 0.1201 | 0.3436 | 0.2596 | 0.2665 | 0.0982 | 0.3053 |
| | DCP | 0.4071 | 0.4414 | 0.2488 | 0.4342 | 0.3958 | 0.4264 | 0.2350 | 0.4265 | 0.3879 | 0.4095 | 0.2359 | 0.4166 | 0.3784 | 0.3849 | 0.2113 | 0.4026 |
| | CPSPAN | 0.3574 | 0.3457 | 0.1911 | 0.4103 | 0.3161 | 0.2783 | 0.1541 | 0.3439 | 0.2553 | 0.2070 | 0.1063 | 0.2786 | 0.2025 | 0.1423 | 0.0621 | 0.2198 |
| | DSIMVC | 0.2795 | 0.2976 | 0.1430 | 0.3218 | 0.2734 | 0.2896 | 0.1383 | 0.3131 | 0.2697 | 0.2828 | 0.1335 | 0.3133 | 0.2654 | 0.2741 | 0.1290 | 0.3062 |
| | DVIMC | 0.4863 | 0.4780 | 0.3192 | 0.5021 | 0.4678 | 0.4440 | 0.3097 | 0.4823 | 0.4371 | 0.4163 | 0.2737 | 0.4577 | 0.4136 | 0.3950 | 0.2519 | 0.4286 |
| | MVP | 0.4451 | 0.4361 | 0.2696 | 0.4830 | 0.4458 | 0.4341 | 0.2723 | 0.4866 | 0.4564 | 0.4314 | 0.2967 | **0.5170** | 0.4407 | **0.4285** | 0.2725 | **0.4895** |
| | PMIMC | 0.3317 | 0.3409 | 0.1778 | 0.3602 | 0.3071 | 0.3175 | 0.1484 | 0.3500 | 0.3125 | 0.3180 | 0.1477 | 0.3513 | 0.3178 | 0.3173 | 0.1492 | 0.3509 |
| | ACOVA (Ours) | **0.5040** | **0.4804** | **0.3273** | **0.5270** | **0.4807** | **0.4560** | **0.3106** | **0.5119** | **0.4710** | **0.4360** | **0.2973** | 0.4987 | **0.4428** | 0.4083 | **0.2733** | 0.4636 |
| **Caltech5v** | BSV | 0.5678 | 0.4364 | 0.3408 | 0.5827 | 0.5035 | 0.3699 | 0.2308 | 0.5170 | 0.4483 | 0.3235 | 0.1488 | 0.4617 | 0.3859 | 0.2798 | 0.0890 | 0.4019 |
| | Concat | 0.4750 | 0.3300 | 0.2519 | 0.4957 | 0.4450 | 0.2941 | 0.1595 | 0.4593 | 0.4279 | 0.2756 | 0.1595 | 0.4386 | 0.3600 | 0.2338 | 0.0891 | 0.3757 |
| | DCP | 0.5754 | 0.5719 | 0.4612 | 0.6054 | 0.5363 | 0.5269 | 0.4199 | 0.5751 | 0.5063 | 0.4729 | 0.3406 | 0.5393 | 0.3982 | 0.3363 | 0.1378 | 0.4056 |
| | CPSPAN | 0.7689 | 0.6677 | 0.6088 | 0.7750 | 0.6874 | 0.5653 | 0.4666 | 0.6888 | 0.6010 | 0.4789 | 0.3467 | 0.6030 | 0.4861 | 0.3664 | 0.2222 | 0.4904 |
| | DSIMVC | 0.7760 | 0.6998 | 0.6280 | 0.7816 | 0.7841 | 0.6947 | 0.6335 | 0.7841 | 0.7339 | 0.6868 | 0.5989 | 0.7413 | 0.7029 | 0.5838 | 0.5083 | 0.7047 |
| | DVIMC | 0.9013 | 0.8207 | 0.7999 | 0.9013 | 0.8921 | 0.8011 | 0.7850 | 0.8921 | 0.8609 | 0.7507 | 0.7322 | 0.8609 | 0.8514 | 0.7345 | 0.7195 | 0.8514 |
| | MVP | 0.8083 | 0.6969 | 0.6594 | 0.8083 | 0.8070 | 0.7139 | 0.6723 | 0.8146 | 0.8103 | 0.7063 | 0.6700 | 0.8157 | 0.7907 | 0.6483 | 0.5934 | 0.7693 |
| | PMIMC | 0.8963 | 0.8205 | 0.7966 | 0.8963 | 0.8917 | 0.8124 | 0.7858 | 0.8917 | 0.8601 | 0.7591 | 0.7381 | 0.8601 | 0.8570 | 0.7522 | 0.7304 | 0.8570 |
| | ACOVA (Ours) | **0.9149** | **0.8507** | **0.8287** | **0.9149** | **0.9131** | **0.8435** | **0.8226** | **0.9131** | **0.8953** | **0.8078** | **0.7918** | **0.8953** | **0.8660** | **0.7531** | **0.7340** | **0.8660** |
| **Handwritten** | BSV | 0.7087 | 0.6647 | 0.5625 | 0.7433 | 0.7105 | 0.6344 | 0.4716 | 0.7171 | 0.5842 | 0.5444 | 0.3028 | 0.5970 | 0.5159 | 0.4802 | 0.1874 | 0.5208 |
| | Concat | 0.7684 | 0.7214 | 0.6412 | 0.7837 | 0.7147 | 0.6441 | 0.5012 | 0.7190 | 0.6108 | 0.5437 | 0.3492 | 0.6166 | 0.5021 | 0.4498 | 0.2511 | 0.5105 |
| | DCP | 0.6459 | 0.7094 | 0.5663 | 0.6737 | 0.6639 | 0.6951 | 0.5189 | 0.6877 | 0.7045 | 0.7018 | 0.5651 | 0.7334 | 0.6913 | 0.6398 | 0.4856 | 0.7083 |
| | CPSPAN | 0.8710 | 0.7969 | 0.7586 | 0.8727 | 0.7524 | 0.6927 | 0.5603 | 0.7638 | 0.6430 | 0.6050 | 0.3862 | 0.6595 | 0.6470 | 0.5828 | 0.3289 | 0.6486 |
| | DSIMVC | 0.8014 | 0.7960 | 0.7282 | 0.8158 | 0.8180 | 0.8082 | 0.7455 | 0.8275 | 0.8304 | 0.8122 | 0.7543 | 0.8342 | 0.8284 | 0.7879 | 0.7312 | 0.8284 |
| | DVIMC | 0.9169 | 0.8604 | 0.8547 | 0.9201 | 0.8954 | 0.8470 | 0.8210 | 0.9020 | 0.8980 | 0.8373 | 0.8175 | 0.9054 | 0.7808 | 0.7905 | 0.7197 | 0.8099 |
| | MVP | 0.7833 | 0.8002 | 0.7114 | 0.8119 | 0.8095 | 0.8000 | 0.7270 | 0.8208 | 0.7474 | 0.7853 | 0.6834 | 0.7715 | 0.7483 | **0.8325** | **0.7758** | 0.7678 |
| | PMIMC | 0.8826 | 0.8188 | 0.7846 | 0.8834 | 0.8998 | 0.8213 | 0.7962 | 0.8998 | 0.8613 | 0.8096 | 0.7637 | 0.8695 | 0.8522 | 0.7994 | 0.7505 | 0.8587 |
| | ACOVA (Ours) | **0.9363** | **0.8692** | **0.8645** | **0.9363** | **0.9376** | **0.8709** | **0.8669** | **0.9376** | **0.9146** | **0.8489** | **0.8325** | **0.9146** | **0.8999** | 0.8351 | 0.8176 | **0.9021** |
| **Fashion** | BSV | 0.4832 | 0.4606 | 0.2986 | 0.5210 | 0.4424 | 0.4114 | 0.2210 | 0.4779 | 0.4100 | 0.3734 | 0.1626 | 0.4361 | 0.3645 | 0.3348 | 0.1040 | 0.3916 |
| | Concat | 0.6546 | 0.6518 | 0.5451 | 0.7028 | 0.5002 | 0.4618 | 0.3646 | 0.5272 | 0.4146 | 0.3433 | 0.2509 | 0.4350 | 0.2988 | 0.1976 | 0.1307 | 0.3247 |
| | DCP | 0.8547 | 0.8864 | 0.8119 | 0.8652 | 0.7663 | 0.8340 | 0.7285 | 0.7899 | 0.7533 | 0.7995 | 0.6904 | 0.7643 | 0.6717 | 0.7430 | 0.6079 | 0.6879 |
| | CPSPAN | 0.6546 | 0.6518 | 0.5451 | 0.7028 | 0.5002 | 0.4618 | 0.3646 | 0.5272 | 0.4146 | 0.3433 | 0.2509 | 0.4350 | 0.2988 | 0.1976 | 0.1307 | 0.3247 |
| | DSIMVC | 0.8993 | 0.8573 | 0.8178 | 0.8993 | 0.8810 | **0.8772** | 0.8162 | 0.8810 | 0.8271 | 0.7990 | 0.7402 | 0.7403 | 0.8064 | 0.7707 | 0.6951 | 0.7904 |
| | DVIMC | 0.8856 | 0.8799 | 0.8284 | 0.9024 | 0.8806 | 0.8718 | 0.8236 | 0.8986 | 0.8629 | 0.8449 | **0.7965** | **0.8867** | **0.8579** | 0.8290 | 0.7729 | 0.8725 |
| | MVP | 0.8691 | 0.8743 | 0.8175 | 0.8955 | 0.8259 | 0.8537 | 0.7744 | 0.8722 | 0.7982 | 0.8372 | 0.7484 | 0.8485 | 0.8386 | 0.8325 | 0.7758 | 0.8665 |
| | PMIMC | 0.6986 | 0.7563 | 0.6282 | 0.7427 | 0.7039 | 0.7615 | 0.6365 | 0.7535 | 0.7118 | 0.7586 | 0.6295 | 0.7577 | 0.7059 | 0.7583 | 0.6362 | 0.7540 |
| | ACOVA (Ours) | **0.9056** | **0.8876** | **0.8516** | **0.9129** | **0.8848** | 0.8735 | **0.8266** | **0.9009** | **0.8764** | **0.8461** | 0.7815 | 0.8736 | 0.8503 | **0.8461** | **0.7815** | **0.8736** |
| **NoisyMNIST** | BSV | 0.5144 | 0.4542 | 0.3258 | 0.5618 | 0.4184 | 0.3343 | 0.1896 | 0.4456 | 0.2939 | 0.1912 | 0.0754 | 0.3090 | 0.2672 | 0.1566 | 0.0452 | 0.2693 |
| | Concat | 0.4338 | 0.3910 | 0.2727 | 0.4418 | 0.3873 | 0.3229 | 0.1891 | 0.4036 | 0.3463 | 0.2592 | 0.1252 | 0.3561 | 0.2805 | 0.1832 | 0.0615 | 0.2864 |
| | DCP | 0.8945 | 0.9149 | 0.8786 | 0.9319 | 0.8997 | 0.8952 | 0.8608 | 0.9198 | 0.9150 | 0.8586 | 0.8401 | 0.9150 | 0.9225 | 0.8335 | 0.8431 | 0.9225 |
| | CPSPAN | 0.5665 | 0.5870 | 0.4402 | 0.6058 | 0.6103 | 0.6050 | 0.4724 | 0.6357 | 0.5774 | 0.5824 | 0.4439 | 0.6097 | 0.5486 | 0.5766 | 0.4271 | 0.5932 |
| | DSIMVC | 0.8993 | 0.8573 | 0.8178 | 0.8977 | 0.7337 | 0.6790 | 0.6148 | 0.7464 | 0.6152 | 0.5769 | 0.4848 | 0.6361 | 0.6245 | 0.5822 | 0.4867 | 0.6343 |
| | DVIMC | 0.9372 | 0.9345 | 0.9157 | 0.9466 | **0.9365** | **0.9060** | **0.8987** | **0.9368** | 0.8998 | 0.8699 | 0.8538 | 0.9019 | 0.8741 | 0.8184 | 0.7999 | 0.8763 |
| | MVP | 0.8657 | 0.8778 | 0.7864 | 0.8667 | 0.8590 | 0.8635 | 0.8222 | 0.8777 | 0.8355 | 0.8195 | 0.7792 | 0.8464 | 0.6660 | 0.6744 | 0.5721 | 0.6694 |
| | PMIMC | 0.7127 | 0.6985 | 0.5927 | 0.7432 | 0.7187 | 0.6593 | 0.5623 | 0.7458 | 0.6616 | 0.5739 | 0.4947 | 0.6695 | 0.6044 | 0.5344 | 0.4393 | 0.6220 |
| | ACOVA (Ours) | **0.9663** | **0.9478** | **0.9440** | **0.9664** | 0.9344 | 0.9125 | 0.9016 | 0.9348 | **0.9599** | **0.9001** | **0.9144** | **0.9599** | **0.9377** | **0.8579** | **0.8697** | **0.9377** |
| **CUB Image-Caption** | BSV | 0.6723 | 0.6604 | 0.4818 | 0.6823 | 0.6080 | 0.5858 | 0.3376 | 0.6163 | 0.5270 | 0.5241 | 0.2295 | 0.5387 | 0.4493 | 0.4474 | 0.1266 | 0.4630 |
| | CONCAT | 0.6890 | 0.6685 | 0.4979 | 0.6947 | 0.5837 | 0.5790 | 0.3322 | 0.5987 | 0.5187 | 0.5107 | 0.2068 | 0.5323 | 0.4663 | 0.4658 | 0.1391 | 0.4770 |
| | DSIMVC | 0.5227 | 0.5822 | 0.5833 | 0.2867 | 0.5163 | 0.5781 | 0.3008 | 0.5797 | 0.4040 | 0.4430 | 0.1196 | 0.4587 | 0.2653 | 0.2337 | 0.0571 | 0.2867 |
| | DCP | 0.6687 | 0.6764 | 0.6356 | 0.5396 | 0.5560 | 0.5603 | 0.5767 | 0.3731 | 0.5497 | 0.5340 | 0.5100 | 0.2823 | 0.4830 | 0.4810 | 0.4547 | 0.1933 |
| | CSPAN | 0.6380 | 0.5744 | 0.4280 | 0.6427 | 0.6420 | 0.5663 | 0.4236 | 0.6420 | 0.5810 | 0.5419 | 0.3824 | 0.5860 | 0.5410 | 0.4965 | 0.3275 | 0.5440 |
| | DVIMC | 0.6573 | 0.6585 | 0.5246 | 0.6807 | 0.6703 | 0.6502 | 0.5131 | 0.6807 | 0.6057 | 0.6026 | 0.4538 | 0.6160 | 0.4710 | 0.4694 | 0.2795 | 0.4803 |
| | MVP | 0.6497 | 0.6317 | 0.5051 | 0.6667 | 0.6840 | 0.6545 | 0.5256 | 0.6903 | 0.6163 | 0.5750 | 0.4401 | 0.6287 | 0.6440 | 0.6036 | 0.4655 | 0.6510 |
| | PMIMC | 0.7183 | 0.7131 | 0.5795 | 0.7383 | 0.6820 | 0.6847 | 0.5360 | 0.6977 | 0.6517 | 0.6198 | 0.4846 | 0.6590 | 0.5907 | 0.5480 | 0.4034 | 0.6060 |
| | ACOVA (Ours) | **0.7970** | **0.7695** | **0.6581** | **0.7970** | **0.7563** | **0.7335** | **0.6215** | **0.7590** | **0.6823** | **0.6568** | **0.5267** | **0.6917** | **0.6554** | **0.6136** | **0.4730** | **0.6584** |

we assume view independence during posterior aggregation. Note, ACOVA in this case simplifies to baseline (Xu et al., 2024). As shown in Table 2, we consistently outperform '*w/o corr.*' across all missing rates on both datasets, indicating the effectiveness and robustness of adaptively modeling and learning view correlations during posterior aggregation. To provide a more comprehensive analysis, we further conduct the grid search over fixed correlation coefficients in [0.1, 0.3, 0.5, 0.7, 0.9]. Note, while the fixed grid search sometimes yields competitive results, its optimal value varies significantly across datasets and missing rates, making it impractical to determine without ground-truth supervision in real clustering scenarios, especially for more severe missing conditions. In contrast, our correlation learning strategy adaptively captures the intrinsic relationships between views without manual tuning or supervision, demonstrating both effectiveness and robustness of our method.

**Effectiveness of Training Objectives.** Our objective function consists of two components: $\mathcal{L}_{\text{ELBO}}$ and $\mathcal{L}_{\text{align}}$. As demonstrated in Table 2. Removing either of them leads to significant performance drops, verifying their complementary roles, where the $\mathcal{L}_{\text{ELBO}}$ ensures effective representation learning and $\mathcal{L}_{\text{align}}$ promotes cross-view consistency.

**Table 2:** Ablation results on Handwritten and Scene15 datasets with different missing-view rates. '*w/*' and '*w/o*' mean '*with*' and '*without*', respectively. 'corr.' refers to considering cross-view correlations. The best result in each column is denoted in **bold**. Note that all results are averaged over five runs.

| | 10% | | | | 30% | | | | 50% | | | | 70% | | | |
|---|---|---|---|---|---|---|---|---|---|---|---|---|---|---|---|---|
| | ACC↑ | NMI↑ | ARI↑ | PUR↑ | ACC↑ | NMI↑ | ARI↑ | PUR↑ | ACC↑ | NMI↑ | ARI↑ | PUR↑ | ACC↑ | NMI↑ | ARI↑ | PUR↑ |
| **Handwritten** | | | | | | | | | | | | | | | | |
| Learn **R** (*w/* corr.) | **0.9363** | 0.8692 | 0.8645 | **0.9363** | **0.9376** | **0.8709** | **0.8669** | **0.9376** | **0.9146** | **0.8489** | **0.8325** | **0.9146** | 0.8999 | **0.8351** | **0.8176** | 0.9021 |
| *w/o* corr. | 0.9169 | 0.8604 | 0.8547 | 0.9201 | 0.8954 | 0.8470 | 0.8210 | 0.9020 | 0.8980 | 0.8373 | 0.8175 | 0.9054 | 0.7808 | 0.7905 | 0.7197 | 0.8099 |
| fixed $\rho = 0.1$ | 0.9339 | 0.8676 | 0.8595 | 0.9339 | 0.9249 | 0.8534 | 0.8418 | 0.9249 | 0.8860 | 0.8288 | 0.8031 | 0.8898 | 0.8881 | 0.8231 | 0.7986 | 0.8918 |
| fixed $\rho = 0.3$ | 0.8874 | 0.8513 | 0.8231 | 0.8945 | 0.9217 | 0.8509 | 0.8355 | 0.9217 | 0.8929 | 0.8335 | 0.8093 | 0.8972 | 0.8859 | 0.8219 | 0.7962 | 0.8903 |
| fixed $\rho = 0.5$ | 0.9088 | 0.8590 | 0.8391 | 0.9118 | 0.9301 | 0.8612 | 0.8510 | 0.9301 | 0.8710 | 0.8291 | 0.7957 | 0.8821 | 0.8867 | 0.8239 | 0.7962 | 0.8908 |
| fixed $\rho = 0.7$ | 0.9069 | 0.8568 | 0.8322 | 0.9091 | 0.9317 | 0.8638 | 0.8545 | 0.9317 | 0.8963 | 0.8410 | 0.8161 | 0.9001 | **0.9115** | 0.8319 | 0.8145 | **0.9115** |
| fixed $\rho = 0.9$ | 0.9362 | **0.8749** | **0.8649** | 0.9362 | 0.9172 | 0.8445 | 0.8263 | 0.9172 | 0.8786 | 0.8330 | 0.8036 | 0.8828 | 0.8977 | 0.8140 | 0.7934 | 0.8977 |
| *w/o* $\mathcal{L}_{\text{ELBO}}$ | 0.6240 | 0.5284 | 0.3673 | 0.6240 | 0.5980 | 0.4548 | 0.3227 | 0.5980 | 0.5495 | 0.3924 | 0.2874 | 0.5545 | 0.4845 | 0.3357 | 0.2301 | 0.5020 |
| *w/o* $\mathcal{L}_{\text{align}}$ | 0.8235 | 0.7768 | 0.7036 | 0.8235 | 0.6930 | 0.6378 | 0.5179 | 0.6930 | 0.6645 | 0.6259 | 0.5179 | 0.6675 | 0.6090 | 0.5664 | 0.4574 | 0.6100 |
| **Scene15** | | | | | | | | | | | | | | | | |
| Learn **R** (*w/* corr.) | 0.5040 | **0.4804** | **0.3273** | 0.5270 | **0.4807** | **0.4560** | **0.3106** | **0.5119** | **0.4710** | **0.4360** | **0.2973** | **0.4987** | 0.4428 | **0.4083** | **0.2733** | **0.4636** |
| *w/o* corr. | 0.4863 | 0.4780 | 0.3192 | 0.5021 | 0.4678 | 0.4440 | 0.3097 | 0.4823 | 0.4371 | 0.4163 | 0.2737 | 0.4577 | 0.4136 | 0.3950 | 0.2519 | 0.4286 |
| fixed $\rho = 0.1$ | 0.4963 | 0.4722 | 0.3237 | 0.5220 | 0.4702 | 0.4466 | 0.2977 | 0.4941 | 0.4653 | 0.4292 | 0.2955 | 0.4966 | 0.4181 | 0.3878 | 0.2518 | 0.4287 |
| fixed $\rho = 0.3$ | **0.5057** | 0.4743 | 0.3272 | **0.5281** | 0.4746 | 0.4458 | 0.3011 | 0.4986 | 0.4598 | 0.4275 | 0.2901 | 0.4885 | 0.4165 | 0.3882 | 0.2508 | 0.4309 |
| fixed $\rho = 0.5$ | 0.4963 | 0.4673 | 0.3214 | 0.5218 | 0.4689 | 0.4414 | 0.3005 | 0.4891 | 0.4569 | 0.4224 | 0.2863 | 0.4815 | 0.4105 | 0.3893 | 0.2521 | 0.4228 |
| fixed $\rho = 0.7$ | 0.4761 | 0.4543 | 0.3023 | 0.5031 | 0.4742 | 0.4255 | 0.2907 | 0.4940 | 0.4590 | 0.4295 | 0.2818 | 0.4765 | 0.4192 | 0.3827 | 0.2503 | 0.4353 |
| fixed $\rho = 0.9$ | 0.4649 | 0.4505 | 0.2938 | 0.4939 | 0.4609 | 0.4187 | 0.2710 | 0.4727 | 0.3918 | 0.3877 | 0.2083 | 0.4144 | 0.3656 | 0.3575 | 0.2239 | 0.3765 |
| *w/o* $\mathcal{L}_{\text{ELBO}}$ | 0.1599 | 0.1605 | 0.0519 | 0.1632 | 0.1407 | 0.0870 | 0.0050 | 0.1431 | 0.1405 | 0.0854 | 0.0220 | 0.1411 | 0.1394 | 0.0746 | 0.0170 | 0.1402 |
| *w/o* $\mathcal{L}_{\text{align}}$ | 0.4566 | 0.4624 | 0.2926 | 0.4941 | 0.3974 | 0.4000 | 0.2335 | 0.4193 | 0.3393 | 0.3412 | 0.1902 | 0.3454 | 0.2105 | 0.1798 | 0.0821 | 0.2157 |

## 5.4 Correlation Learning Analysis

To further demonstrate the view relationships captured by the correlation matrix **R** and validate our motivation, we conducted controlled simulation experiments on the Fashion dataset. Specifically, we created two views by duplicating the original images. To simulate heterogeneous views, we adopted a mutually exclusive noise injection strategy where each sample had exactly one corrupted view (view 1 for randomly selected 50% of samples, view 2 for the rest). We introduced pepper noise rates $\in \{0.00, 0.50, 1.00\}$ to control the degree of view heterogeneity, and missing rates $\in \{0.00, 0.10, 0.30, 0.50, 0.70\}$ to simulate incomplete data scenarios.

Figure 3 visualizes the learned off-diagonal correlations in **R** across all combinations of noise and missing rates. A clear bi-directional declining trend emerges: as the noise rate increases from 0.00 to 1.00, the two views gradually become less correlated due to increasing heterogeneity. Similarly, as the missing rate increases, the correlation values in **R** progressively decrease across all noise conditions, as incomplete information limits the shared structure between views. These results validate that our learned **R** adaptively captures meaningful cross-view relationships under both heterogeneous and incomplete conditions, supporting the necessity of adaptive correlation learning for realistic multi-view clustering. Notably, even under maximal noise and high

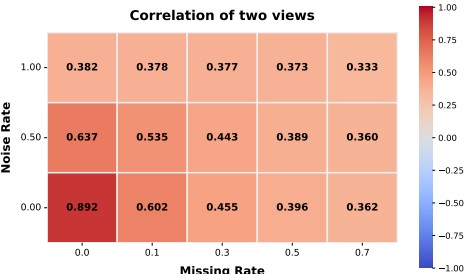

**Figure 3:** Learned correlations under different noise and missing rates.

missingness, the off-diagonal correlations do not collapse fully to zero (the prior). We hypothesize that this arises from the generative model itself, which requires the latent variables to retain some meaningful information to generate the views.

## 5.5 Visualization and Parameter Sensitivity Analysis

**Visualization.** Figure 4(a-c) compares the t-SNE visualization between CONCAT (Zhao et al., 2016), DVIMC (Xu et al., 2024) and our method with a missing rate of 10%. Our approach shows clearer separation and fewer outliers, indicating more discriminative representations by learning cross-view correlations adaptively from incomplete data. In addition, we plot the correlation matrix **R** learned from the incomplete Handwritten datasets with 50% missing rates in Figure 4(d), where we observe meaningful view dependencies. The views for this dataset corresponding to: Fourier coefficients, Profile correlations, Karhunen-Loève coefficients, Zernike moments, Pixel averages, and morphological features (in that order). From Figure 4(d), we notice that most views have the least correlation with the last view, which intuitively makes sense, as the morphological features are

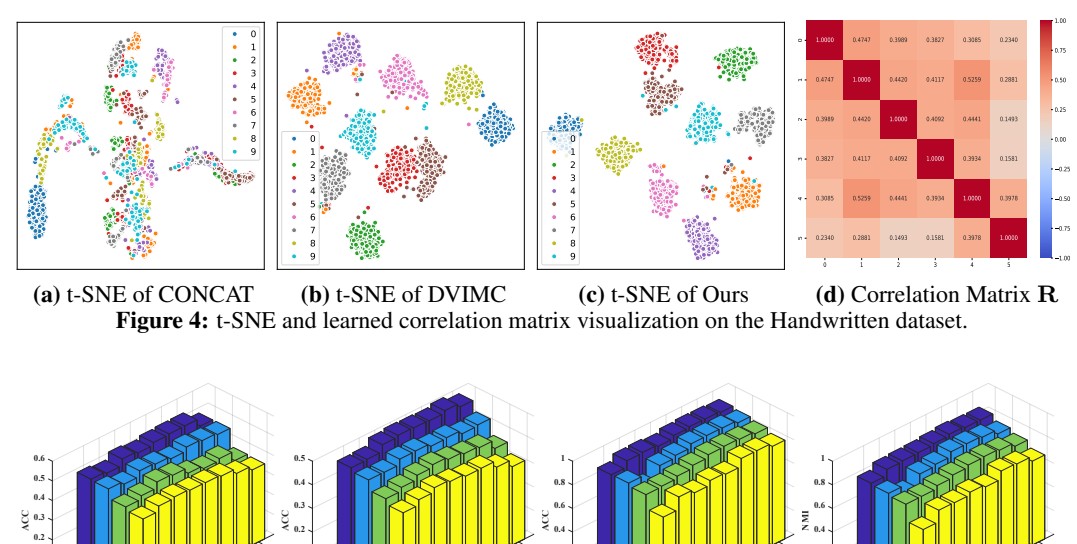

**(a)** t-SNE of CONCAT    **(b)** t-SNE of DVIMC    **(c)** t-SNE of Ours    **(d)** Correlation Matrix $\mathbf{R}$

**Figure 4:** t-SNE and learned correlation matrix visualization on the Handwritten dataset.

**(a)** ACC on Scene15    **(b)** NMI on Scene15    **(c)** ACC on Caltech5V    **(d)** NMI on Caltech5V

**Figure 5:** Clustering performance with different parameter $\alpha$ and missing rate settings on Scene15 and Caltech5V dataset.

significantly different from the frequency- or intensity-based features in the other views. Further, we notice high correlation between profile correlations and pixel averages, where both are linked to the underlying intensity distributions. We also observe high correlation between Karhunen-Loève coefficients and pixel averages, which indicates that much of the variation in the dataset is due to average intensities, which also aligns well with what we expect for this dataset. More visualization on different datasets and missing rates are included in Appendix C.

**Parameter Sensitivity Analysis.** This section empirically analyzes the effect on ACC and NMI of the trade-off parameter $\alpha$ in Eq. 15 across different missing rates for the Scene15 and Caltech5V datasets. As shown in Figure 5, $\alpha$ in the range of [5,20] achieves higher performance.

Additional sensitivity studies and analyses can be found in Appendix C.

## 6 DISCUSSION AND CONCLUSION

In this work, we propose ACOVA, an imputation-free variational framework that adaptively learns inter-view correlations for IMVC. By breaking the conventional independence assumption and modeling cross-view correlation through view-specific estimation error, our method captures the cross-view dependencies. Besides, we propose to jointly optimize the posterior representations and cross-view correlation, enabling more effective latent embedding and significantly boosting clustering performance, especially under high missing-view conditions. Extensive experiments on multiple benchmarks verify the effectiveness and robustness of our approach compared to prior IMVC methods.

## REPRODUCIBILITY STATEMENT

To ensure full reproducibility, we provide the source code in the supplementary material as well as all implementation and dataset details in Sec. 5.1 and Appendix B. We further include proofs of all theoretic results in Appendix A.

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

# Appendices

# A   THEORETICAL DERIVATIONS AND ANALYSIS

## A.1   MATHEMATICAL BACKGROUND

This section provides the mathematical foundations for the key equations in variational incomplete multi-view clustering preliminaries in Section 3.

### A.1.1   DERIVATION OF THE JOINT PROBABILITY (EQUATION 1)

The generative process assumes a probabilistic model where cluster assignment $\mathbf{c}$ is drawn from a categorical distribution, shared latent representation $\mathbf{z}$ depends on the cluster assignment, and each view $\mathbf{x}_v$ is conditionally independent of other views given $\mathbf{z}$. For the joint probability $p(\{\mathbf{x}_v\}_{v=1}^V, \mathbf{z}, \mathbf{c})$, we could first apply the chain rule of probability to get:

$$p(\{\mathbf{x}_v\}_{v=1}^V, \mathbf{z}, \mathbf{c}) = p(\mathbf{c}) \cdot p(\mathbf{z} \mid \mathbf{c}) \cdot p(\{\mathbf{x}_v\}_{v=1}^V \mid \mathbf{z}, \mathbf{c}). \tag{17}$$

Given the latent representation $\mathbf{z}$, we assume the observations are independent of the cluster assignment: $\mathbf{x}_v \perp \mathbf{c}|\mathbf{z}$, which gives $p(\{\mathbf{x}_v\}_{v=1}^V \mid \mathbf{z}, \mathbf{c}) = p(\{\mathbf{x}_v\}_{v=1}^V \mid \mathbf{z})$. Besides, given the shared representation $\mathbf{z}$, we assume different views are conditionally independent: $\mathbf{x}_i \perp \mathbf{x}_j|\mathbf{z}$ for $i \neq j$, yielding $p(\{\mathbf{x}_v\}_{v=1}^V \mid \mathbf{z}) = \prod_{v=1}^V p(\mathbf{x}_v \mid \mathbf{z})$. This leads directly to Eq.1:

$$p(\{\mathbf{x}_v\}_{v=1}^V, \mathbf{z}, \mathbf{c}) = p(\mathbf{c})\, p(\mathbf{z} \mid \mathbf{c}) \prod_{v=1}^V p(\mathbf{x}_v \mid \mathbf{z}). \tag{18}$$

### A.1.2   DERIVATION OF THE POSTERIOR DISTRIBUTION (EQUATION 2)

For incomplete multi-view data where only views $v \in \mathcal{V}$ are observed, the posterior distribution follows from Bayes' theorem:

$$p(\mathbf{z}, \mathbf{c}|\{\mathbf{x}_v\}_{v\in\mathcal{V}}) = \frac{p(\{\mathbf{x}_v\}_{v\in\mathcal{V}}, \mathbf{z}, \mathbf{c})}{p(\{\mathbf{x}_v\}_{v\in\mathcal{V}})}. \tag{19}$$

For the incomplete setting, missing views are implicitly marginalized out. Under the conditional independence assumption, the joint likelihood for observed views becomes:

$$p(\{\mathbf{x}_v\}_{v\in\mathcal{V}} \mid \mathbf{z}) = \prod_{v\in\mathcal{V}} p(\mathbf{x}_v \mid \mathbf{z}). \tag{20}$$

The marginal likelihood requires summing over all latent variables, yielding Eq.2:

$$p(\mathbf{z}, \mathbf{c}|\{\mathbf{x}_v\}_{v\in\mathcal{V}}) = \frac{p(\{\mathbf{x}_v\}_{v\in\mathcal{V}}|\mathbf{z})p(\mathbf{z}, \mathbf{c})}{\int_{\mathbf{z}} \sum_{\mathbf{c}=1}^C p(\{\mathbf{x}_v\}_{v\in\mathcal{V}}|\mathbf{z}, \mathbf{c})p(\mathbf{z}, \mathbf{c})d\mathbf{z}}. \tag{21}$$

### A.1.3   VARIATIONAL APPROXIMATION (EQUATION 3)

The exact posterior is computationally intractable due to the complex integration and summation in the denominator. Variational inference approximates this with a tractable distribution $q_\phi(\mathbf{z}, \mathbf{c}|\{\mathbf{x}_v\}_{v\in\mathcal{V}})$ by minimizing the Kullback-Leibler divergence (Eq.3):

$$D_{\mathrm{KL}}(q_\phi(\mathbf{z}, \mathbf{c}|\{\mathbf{x}_v\}_{v\in\mathcal{V}})||p(\mathbf{z}, \mathbf{c}|\{\mathbf{x}_v\}_{v\in\mathcal{V}})). \tag{22}$$

As for optimization, minimizing the KL divergence is equivalent to maximizing the Evidence Lower BOund (ELBO), and please refer to A.5 for more detailed derivations.

## A.2   PROOF OF THE FROBENIUS IDENTITY DEVIATION BOUND

We provide the proof of the Theorem of Frobenius Identity Deviation Bound, which establishes an upper bound on the Frobenius norm deviation of a correlation matrix from the identity.

### A.3 PROOF OF IDENTIFIABILITY OF THE COVARIANCE PARAMETERIZATION

In the following, we demonstrate the identifiability of the proposed covariance parameterization.

**Theorem A.1** (Identifiability of the Covariance Parameterization). *Let $\mathbf{D} = \mathrm{diag}(\boldsymbol{\sigma}_1, \ldots, \boldsymbol{\sigma}_V)$ and $\mathbf{D}' = \mathrm{diag}(\sigma'_1, \ldots, \sigma'_V)$ be diagonal matrices with strictly positive entries. Let $\mathbf{L}$ and $\mathbf{L}'$ be lower–triangular matrices with strictly positive diagonal entries, and define correlation matrices*

$$\mathbf{R} = \mathrm{corr}(\mathbf{L}\mathbf{L}^\top), \qquad \mathbf{R}' = \mathrm{corr}(\mathbf{L}'\mathbf{L}'^\top),$$

*i.e. $\mathbf{R}_{ij} = (\mathbf{L}\mathbf{L}^\top)_{ij}/\sqrt{(\mathbf{L}\mathbf{L}^\top)_{ii}(\mathbf{L}\mathbf{L}^\top)_{jj}}$ (and analogously for $\mathbf{R}'$). For the $d$-th dimension, define the covariance matrices*

$$\boldsymbol{\Sigma}_d = \mathbf{D}\mathbf{R}\mathbf{D}, \qquad \boldsymbol{\Sigma}'_d = \mathbf{D}'\mathbf{R}'\mathbf{D}'.$$

*If $\boldsymbol{\Sigma}_d = \boldsymbol{\Sigma}'_d$, then*

$$\mathbf{D} = \mathbf{D}', \qquad \mathbf{R} = \mathbf{R}', \qquad \mathbf{L} = \mathbf{L}'.$$

*In particular, the mapping $(\mathbf{L}, \mathbf{D}) \mapsto \boldsymbol{\Sigma}_d$ is identifiable.*

*Proof.* Assume $\boldsymbol{\Sigma}_d = \boldsymbol{\Sigma}'_d$, i.e.

$$\mathbf{D}\mathbf{R}\mathbf{D} = \mathbf{D}'\mathbf{R}'\mathbf{D}'. \tag{23}$$

Since $\mathbf{D}$ is invertible, left- and right-multiplying Eq. 23 by $\mathbf{D}^{-1}$ yields

$$\mathbf{R} = \mathbf{A}\mathbf{R}'\mathbf{A}, \qquad \mathbf{A} := \mathbf{D}^{-1}\mathbf{D}'.$$

Here $\mathbf{A} = \mathrm{diag}(\mathbf{a}_1, \ldots, \mathbf{a}_V)$ with all $\mathbf{a}_i > 0$. Because $\mathbf{R}$ and $\mathbf{R}'$ are correlation matrices, we have $\mathbf{R}_{ii} = \mathbf{R}'_{ii} = 1$ for all $i$. Taking the diagonal entries of the equation $\mathbf{R} = \mathbf{A}\mathbf{R}'\mathbf{A}$ gives

$$1 = \mathbf{R}_{ii} = (\mathbf{A}\mathbf{R}'\mathbf{A})_{ii} = \mathbf{a}_i^2 \mathbf{R}'_{ii} = \mathbf{a}_i^2, \qquad i = 1, \ldots, V.$$

Hence $\mathbf{a}_i = 1$ for all $i$, implying $\mathbf{A} = \mathbf{I}$ and thus $\mathbf{D}' = \mathbf{D}$. Substituting $\mathbf{D}' = \mathbf{D}$ into Eq. 23 gives

$$\mathbf{D}\mathbf{R}\mathbf{D} = \mathbf{D}\mathbf{R}'\mathbf{D} \quad \Rightarrow \quad \mathbf{R} = \mathbf{R}'.$$

By construction, $\mathbf{R}$ and $\mathbf{R}'$ are correlation matrices obtained from $\mathbf{L}\mathbf{L}^\top$ and $\mathbf{L}'\mathbf{L}'^\top$, respectively. Equality $\mathbf{R} = \mathbf{R}'$ implies $\mathbf{L}\mathbf{L}^\top = \mathbf{L}'\mathbf{L}'^\top$. For any positive definite matrix, the Cholesky factor that is lower triangular with strictly positive diagonal entries is unique. Therefore, $\mathbf{L} = \mathbf{L}'$. Thus $\boldsymbol{\Sigma}_d = \boldsymbol{\Sigma}'_d$ implies $\mathbf{D} = \mathbf{D}'$, $\mathbf{R} = \mathbf{R}'$, and $\mathbf{L} = \mathbf{L}'$, proving identifiability of the parameterization. $\square$

Note, the only non-identifiable quantities are the internal neural-network weights themselves, which is expected and does not affect the identifiability of the covariance structure.

### A.4 COMPLETE DERIVATION OF POSTERIOR DISTRIBUTION

In this section, we provide a detailed derivation of the aggregated posterior distribution under incomplete multi-view settings. Specifically, given a subset of available modalities indexed by a binary mask $\mathbf{M} \in \{0,1\}^V$, we derive the variational posterior $q(\mathbf{z} \mid \{\mathbf{x}_v\}_{v \in \mathcal{V}})$. Note that since our inference model produces view-specific latent predictions $\boldsymbol{\mu}$, this posterior can be equivalently expressed as $q(\mathbf{z}|\boldsymbol{\mu}, \mathbf{M})$, where $\boldsymbol{\mu}$ contains the sufficient statistics from the encoder networks.

$$q(\mathbf{z} \mid \{\mathbf{x}_v\}_{v \in \mathcal{V}}) = q(\mathbf{z}|\boldsymbol{\mu}, \mathbf{M}) \sim \mathcal{N}(\tilde{\boldsymbol{\mu}}, \tilde{\boldsymbol{\sigma}}^2) = \mathcal{N}(\mathbf{A}_\mathbf{M}^{-1}\mathbf{B}_\mathbf{M}, \mathbf{A}_\mathbf{M}^{-1}) \tag{24}$$

We begin by modeling the latent estimation as:

$$\boldsymbol{\mu} = \mathbb{1} \cdot \mathbf{z} + \boldsymbol{\epsilon}, \quad \boldsymbol{\epsilon} \sim \mathcal{N}(\mathbf{0}, \boldsymbol{\Sigma}), \tag{25}$$

where $\boldsymbol{\mu} \in \mathbb{R}^V$ is the vector of view-specific latent predictions, $\mathbf{z} \in \mathbb{R}^d$ is the shared latent variable, $\boldsymbol{\Sigma} \in \mathbb{R}^{V \times V}$ is the correlated noise covariance. Here, $\mathbb{1}$ is a binary design matrix to enable dimensional alignment. The expectation and covariance of $\boldsymbol{\mu}$ are thus:

$$\mathbb{E}[\boldsymbol{\mu}] = \mathbb{E}[\mathbb{1} \cdot \mathbf{z} + \boldsymbol{\epsilon}] = \mathbb{1} \cdot \mathbf{z},$$
$$\mathrm{Cov}[\boldsymbol{\mu}] = \mathrm{Cov}[\boldsymbol{\epsilon}] = \boldsymbol{\Sigma} \tag{26}$$

That is,

$$\boldsymbol{\mu} \sim \mathcal{N}(\mathbb{1} \cdot \mathbf{z}, \boldsymbol{\Sigma}) \tag{27}$$

When only a subset of views are observed, represented by mask $\mathbf{M}$, we define masked observations as:

$$\begin{aligned} \boldsymbol{\mu}_{\mathbf{M}} &= \boldsymbol{\mu} \odot \mathbf{M}, \\ \boldsymbol{\mu}_{\mathbf{M}} &\sim \mathcal{N}(\mathbb{1} \cdot \mathbf{z} \odot \mathbf{M}, \boldsymbol{\Sigma}_{\mathbf{M}}), \end{aligned} \tag{28}$$

where the masked covariance is given by:

$$\boldsymbol{\Sigma}_{\mathbf{M}} = \boldsymbol{\Sigma} \odot (\mathbf{M}\mathbf{M}^{\top}) \tag{29}$$

Assuming an improper flat prior distribution on $\mathbf{z}$, the variational posterior is proportional to the likelihood function. Therefore,

$$\begin{aligned} q(\mathbf{z}|\boldsymbol{\mu}, \mathbf{M}) &\propto \exp\left[-\frac{1}{2}(\boldsymbol{\mu} \odot \mathbf{M} - \mathbb{1}\mathbf{z})^{\top}(\boldsymbol{\Sigma}^{-1} \odot \mathbf{M}\mathbf{M}^{\top})(\boldsymbol{\mu} \odot \mathbf{M} - \mathbb{1}\mathbf{z})\right] \\ &= \exp\left[-\frac{1}{2}[(\boldsymbol{\mu} \odot \mathbf{M})^{\top}(\boldsymbol{\Sigma}^{-1} \odot \mathbf{M}\mathbf{M}^{\top})(\boldsymbol{\mu} \odot \mathbf{M}) - 2(\boldsymbol{\mu} \odot \mathbf{M})^{\top}(\boldsymbol{\Sigma}^{-1} \odot \mathbf{M}\mathbf{M}^{\top})\mathbb{1}\mathbf{z}\right. \\ &\qquad \left. + \mathbf{z}^{\top}\mathbb{1}^{\top}(\boldsymbol{\Sigma}^{-1} \odot \mathbf{M}\mathbf{M}^{\top})\mathbb{1}\mathbf{z}]\right] \\ &= \exp\left[-\frac{1}{2}[\mathbf{z}^{\top}\mathbf{A}_{\mathbf{M}}\mathbf{z} - 2\mathbf{z}^{\top}\mathbf{B}_{\mathbf{M}} + \mathbf{C}]\right] \\ &= \exp\left[-\frac{1}{2}[(\mathbf{z} - \mathbf{A}_{\mathbf{M}}^{-1}\mathbf{B}_{\mathbf{M}})^{\top}\mathbf{A}_{\mathbf{M}}(\mathbf{z} - \mathbf{A}_{\mathbf{M}}^{-1}\mathbf{B}_{\mathbf{M}}) - \mathbf{B}_{\mathbf{M}}^{\top}\mathbf{A}_{\mathbf{M}}^{-1}\mathbf{B}_{\mathbf{M}} + \mathbf{C}]\right] \\ &\propto \exp\left[-\frac{1}{2}(\mathbf{z} - \mathbf{A}_{\mathbf{M}}^{-1}\mathbf{B}_{\mathbf{M}})^{\top}\mathbf{A}_{\mathbf{M}}(\mathbf{z} - \mathbf{A}_{\mathbf{M}}^{-1}\mathbf{B}_{\mathbf{M}})\right], \end{aligned} \tag{30}$$

where

$$\begin{aligned} \mathbf{A}_{\mathbf{M}} &= \mathbb{1}^{\top}(\boldsymbol{\Sigma}^{-1} \odot \mathbf{M}\mathbf{M}^{\top})\mathbb{1} \\ \mathbf{B}_{\mathbf{M}} &= \mathbb{1}^{\top}(\boldsymbol{\Sigma}^{-1} \odot \mathbf{M}\mathbf{M}^{\top})(\boldsymbol{\mu} \odot \mathbf{M}) \end{aligned} \tag{31}$$

The gradient and Hessian of $\log q(\mathbf{z}|\boldsymbol{\mu}, \mathbf{M})$ with respect to $\mathbf{z}$ are:

$$\begin{aligned} \nabla_{\mathbf{z}} \log q(\mathbf{z}|\boldsymbol{\mu}, \mathbf{M}) &= -\mathbf{A}_{\mathbf{M}}\mathbf{z} + \mathbf{B}_{\mathbf{M}} \\ \nabla_{\mathbf{z}}^2 \log q(\mathbf{z}|\boldsymbol{\mu}, \mathbf{M}) &= -\mathbf{A}_{\mathbf{M}} \end{aligned} \tag{32}$$

Therefore, we obtain the final form of the aggregated variational posterior distribution:

$$q(\mathbf{z} \mid \{\mathbf{x}_v\}_{v \in \mathcal{V}}) = q(\mathbf{z}|\boldsymbol{\mu}, \mathbf{M}) \sim \mathcal{N}(\mathbf{A}_{\mathbf{M}}^{-1}\mathbf{B}_{\mathbf{M}}, \mathbf{A}_{\mathbf{M}}^{-1}) \qquad \square$$

## A.5 COMPLETE DERIVATION OF ELBO

In this section, we provide a detailed derivation of the evidence lower bound (ELBO), *i.e.*:

$$\begin{aligned} \mathcal{L}_{\text{ELBO}}(\{\mathbf{x}_v\}_{v \in \mathcal{V}}) &= \mathbb{E}_{q_{\phi}(\mathbf{z}|\{\mathbf{x}_v\}_{v \in \mathcal{V}})}\left[\sum_{v \in \mathcal{V}} \log p(\mathbf{x}_v \mid \mathbf{z})\right] \\ &\quad - \mathbb{E}_{q_{\phi}(\mathbf{c}|\{\mathbf{x}_v\}_{v \in \mathcal{V}})}[D_{\text{KL}}(q_{\phi}(\mathbf{z} \mid \{\mathbf{x}_v\}_{v \in \mathcal{V}}) \| p(\mathbf{z} \mid \mathbf{c}))] - D_{\text{KL}}(q_{\phi}(\mathbf{c} \mid \{\mathbf{x}_v\}_{v \in \mathcal{V}}) \| p(\mathbf{c})) \end{aligned} \tag{33}$$

We begin with the Kullback-Leibler (KL) divergence between the variational joint distribution $q_{\phi}(\mathbf{z}, \mathbf{c}|\{\mathbf{x}_v\}_{v \in \mathcal{V}})$ and the true joint posterior $p(\mathbf{z}, \mathbf{c}|\{\mathbf{x}_v\}_{v \in \mathcal{V}})$:

$$D_{\mathrm{KL}}(q_\phi(\mathbf{z}, \mathbf{c}|\{\mathbf{x}_v\}_{v\in\mathcal{V}})\|p(\mathbf{z}, \mathbf{c}|\{\mathbf{x}_v\}_{v\in\mathcal{V}}))$$

$$= \mathbb{E}_{q_\phi(\mathbf{z}, \mathbf{c}|\{\mathbf{x}_v\}_{v\in\mathcal{V}})}\left[\log \frac{q_\phi(\mathbf{z}, \mathbf{c}|\{\mathbf{x}_v\}_{v\in\mathcal{V}})}{p(\mathbf{z}, \mathbf{c}|\{\mathbf{x}_v\}_{v\in\mathcal{V}})}\right]$$

$$= \mathbb{E}_{q_\phi(\mathbf{z}, \mathbf{c}|\{\mathbf{x}_v\}_{v\in\mathcal{V}})}\left[\log \frac{q_\phi(\mathbf{z}, \mathbf{c}|\{\mathbf{x}_v\}_{v\in\mathcal{V}}) \cdot p(\{\mathbf{x}_v\}_{v\in\mathcal{V}})}{p(\mathbf{z}, \mathbf{c}, \{\mathbf{x}_v\}_{v\in\mathcal{V}})}\right]$$

$$= \mathbb{E}_{q_\phi(\mathbf{z}, \mathbf{c}|\{\mathbf{x}_v\}_{v\in\mathcal{V}})}\left[\log \frac{q_\phi(\mathbf{z}, \mathbf{c}|\{\mathbf{x}_v\}_{v\in\mathcal{V}})}{p(\mathbf{z}, \mathbf{c}, \{\mathbf{x}_v\}_{v\in\mathcal{V}})}\right] + \log p(\{\mathbf{x}_v\}_{v\in\mathcal{V}}) \tag{34}$$

Since KL divergence is always non-negative, we have:

$$\log p(\{\mathbf{x}_v\}_{v\in\mathcal{V}}) \geq \mathbb{E}_{q_\phi(\mathbf{z}, \mathbf{c}|\{\mathbf{x}_v\}_{v\in\mathcal{V}})}\left[\log \frac{p(\mathbf{z}, \mathbf{c}, \{\mathbf{x}_v\}_{v\in\mathcal{V}})}{q_\phi(\mathbf{z}, \mathbf{c}|\{\mathbf{x}_v\}_{v\in\mathcal{V}})}\right]$$

$$= \mathcal{L}_{\mathrm{ELBO}}(\{\mathbf{x}_v\}_{v\in\mathcal{V}}) \tag{35}$$

Given the factorization $q_\phi(\mathbf{z}, \mathbf{c}|\{\mathbf{x}_v\}_{v\in\mathcal{V}}) = q_\phi(\mathbf{z}|\{\mathbf{x}_v\}_{v\in\mathcal{V}})q_\phi(\mathbf{c}|\{\mathbf{x}_v\}_{v\in\mathcal{V}})$ and the joint distribution:

$$p(\mathbf{z}, \mathbf{c}, \{\mathbf{x}_v\}_{v\in\mathcal{V}}) = p(\mathbf{c})p(\mathbf{z}|\mathbf{c})\prod_{v\in\mathcal{V}} p(\mathbf{x}_v|\mathbf{z}), \tag{36}$$

we can then derive the Evidence Lower Bound (ELBO):

$$\mathcal{L}_{\mathrm{ELBO}}(\{\mathbf{x}_v\}_{v\in\mathcal{V}})$$

$$= \mathbb{E}_{q_\phi(\mathbf{z}, \mathbf{c}|\{\mathbf{x}_v\}_{v\in\mathcal{V}})}\left[\log \frac{p(\mathbf{z}, \mathbf{c}, \{\mathbf{x}_v\}_{v\in\mathcal{V}})}{q_\phi(\mathbf{z}, \mathbf{c}|\{\mathbf{x}_v\}_{v\in\mathcal{V}})}\right]$$

$$= \mathbb{E}_{q_\phi(\mathbf{z}, \mathbf{c}|\{\mathbf{x}_v\}_{v\in\mathcal{V}})}\left[\log \frac{p(\mathbf{c})p(\mathbf{z}|\mathbf{c})\prod_{v\in\mathcal{V}} p(\mathbf{x}_v|\mathbf{z})}{q_\phi(\mathbf{z}|\{\mathbf{x}_v\}_{v\in\mathcal{V}})q_\phi(\mathbf{c}|\{\mathbf{x}_v\}_{v\in\mathcal{V}})}\right]$$

$$= \mathbb{E}_{q_\phi(\mathbf{z}, \mathbf{c}|\{\mathbf{x}_v\}_{v\in\mathcal{V}})}\left[\sum_{v\in\mathcal{V}} \log p(\mathbf{x}_v|\mathbf{z}) + \log p(\mathbf{c})\right.$$

$$\left.+ \log p(\mathbf{z}|\mathbf{c}) - \log q_\phi(\mathbf{z}|\{\mathbf{x}_v\}_{v\in\mathcal{V}}) - \log q_\phi(\mathbf{c}|\{\mathbf{x}_v\}_{v\in\mathcal{V}})\right]$$

$$= \mathbb{E}_{q_\phi(\mathbf{z}|\{\mathbf{x}_v\}_{v\in\mathcal{V}})}\left[\sum_{v\in\mathcal{V}} \log p(\mathbf{x}_v|\mathbf{z})\right]$$

$$- \mathbb{E}_{q_\phi(\mathbf{c}|\{\mathbf{x}_v\}_{v\in\mathcal{V}})}[D_{\mathrm{KL}}(q_\phi(\mathbf{z}|\{\mathbf{x}_v\}_{v\in\mathcal{V}})\|p(\mathbf{z}|\mathbf{c}))] - D_{\mathrm{KL}}(q_\phi(\mathbf{c}|\{\mathbf{x}_v\}_{v\in\mathcal{V}})\|p(\mathbf{c})) \qquad \square$$

### A.6 COMPLEXITY ANALYSIS

In our framework, the primary computational operations include feature encoding, cross-view aggregation, and posterior inference. Let $N$ be the number of samples, $V$ the number of views. For simplicity, we assume that all views are mapped to a common latent dimension $D$ after encoding, regardless of their original dimensionality $d_v$. The view-specific feature encoding process thus takes $\mathcal{O}(NVD)$. The cross-view correlation modeling, which is the core of our method, involves two steps: (1) constructing $V \times V$ covariance matrices for each sample and latent dimension with complexity $\mathcal{O}(NDV^2)$, and (2) performing matrix inversion on these covariance matrices, which dominates the computational cost with $\mathcal{O}(NDV^3)$. Besides, other reconstruction and KL divergence terms in posterior inference cost $\mathcal{O}(NVD)$. Therefore, the overall time complexity is $\mathcal{O}(NDV^3)$. Since $V$ is usually small (*e.g.*, $V < 10$ and $V \ll N$), our method remains highly efficient in practical incomplete multi-view clustering scenarios, scaling linearly with the number of samples $N$. See Appendix C.3 for a empirical comparison of run-times and Appendix A.7 for a discussion on how the implementation of the proposed algorithm can further be improved when facing large number of views $V$.

### A.7 NOTE ON EFFICIENT AND STABLE INVERSION

Our formulation expresses $\mathbf{\Sigma}^d = \mathbf{DRD}$, with $\mathbf{R}_{ij} = \frac{(\mathbf{LL}^\top)_{ij}}{\sqrt{(\mathbf{LL}^\top)_{ii}(\mathbf{LL}^\top)_{jj}}}$, where $\mathbf{L}$ is a learned lower–triangular matrix with strictly positive diagonal entries. This parameterization guarantees $R \succ 0$. However, while we have not encountered this in our experiments, directly forming or inverting $R^{-1}$ or $\Sigma^{-1}$ could in certain cases be numerically unstable. We therefore outline an alternative that avoids explicitly computing the inverse by exploiting the structure of the Cholesky decomposition.

Given our learned $\mathbf{L}$, $\mathbf{R} = \mathbf{D}_R^{-1}(\mathbf{LL}^T)\mathbf{D}_R^{-1}$ and the inverse $\mathbf{R}^{-1} = \mathbf{D}_R(\mathbf{LL}^T)^{-1}\mathbf{D}_R$, where $\mathbf{D}_R = \mathrm{diag}\left(\sqrt{(\mathbf{LL}^\top)_{11}}, \ldots, \sqrt{(\mathbf{LL}^\top)_{VV}}\right)$. Thus $\mathbf{\Sigma}^{d^{-1}} = \mathbf{D}^{-1}\mathbf{RD}^{-1} = (\mathbf{D}^{-1}\mathbf{D}_R)\mathbf{L}^{-T}\mathbf{L}^{-1}(\mathbf{D}_R\mathbf{D}^{-1})$, where $\mathbf{L}^{-T}$ and $\mathbf{L}^{-1}$ can be computed via forward and backward substitution instead of matrix inversion, being numerically more stable as well as being $\mathcal{O}(V^2)$ instead of $\mathcal{O}(V^3)$.

## B EXPERIMENT SETTINGS

### B.1 STATISTICS FOR DATASETS

In this section, we provide details of the six real-world datasets used in the experiment in Table 3.

**Table 3:** Detailed statistics for the six multi-view clustering datasets used in our experiments.

| Dataset | #Views | #View Dimensions | #Instances | #Categories |
|---|---|---|---|---|
| Handwritten (LeCun et al., 1989) | 6 | 76, 216, 64, 47, 240, 6 | 2,000 | 10 |
| Caltech5V (Fei-Fei et al., 2004) | 5 | 40, 254, 1984, 512, 928 | 1,400 | 7 |
| Scene15 (Fei-Fei & Perona, 2005) | 3 | 20, 59, 40 | 4,485 | 15 |
| Fashion-MV (Xiao et al., 2017) | 3 | 784, 784, 784 | 10,000 | 10 |
| NoisyMNIST (Wang et al., 2015) | 2 | 784, 784 | 70,000 | 10 |
| CUB Image-Captions (Zhang et al., 2019) | 2 | 1024, 300 | 600 | 10 |

### B.2 INCOMPLETE MULTI-VIEW DATA GENERATION

Following prior IMVC works (Xu et al., 2024; Tang & Liu, 2022), we generate incomplete multi-view data by randomly masking a proportion $p\%$ of the samples. Formally, given a dataset with $N$ samples and $V$ views, we define a binary indicator matrix $\mathbf{M} \in \{0,1\}^{N \times V}$, where $\mathbf{M}_{i,v} = 1$ if the $i$-th sample is observed in view $v$, and $\mathbf{M}_{i,v} = 0$ otherwise. The generation process is as follows: (1) randomly select $p\%$ of the samples as incomplete; (2) for each selected sample, randomly assign a non-empty proper subset of views to be retained, ensuring that at least one view is observed and at least one is missing; (3) all remaining $(1-p)\%$ of the samples are kept fully observed with $\mathbf{M}_{i,v} = 1$ for all $v \in \{1, \ldots, V\}$. For example, in a three-view setting with views $\{A, B, C\}$, incomplete samples may have only one view (e.g., $A$, $B$, or $C$) or two views (e.g., $\{A, B\}$, $\{B, C\}$, or $\{A, C\}$).

### B.3 COMPARISON METHODS

In this section, we provide details on the eight comparison approaches that are considered in the experiments (see Table 5).

### B.4 IMPLEMENTATION DETAILS

All experiments are conducted on the PyTorch platform, running on a Linux server equipped with an Intel(R) Xeon(R) Gold 5218R CPU @ 2.10GHz and an NVIDIA GeForce RTX 3090 GPU. For fair comparison, we run five experimental trials for each method and report the average results. All compared methods are implemented with their recommended hyperparameter settings or through a parameter search for better performance. For network architecture, each view-specific encoder consists of three fully-connected layers with dimensions [500, 500, 2,000], followed by ReLU activation functions, while the dimensions of the decoder are reversed. The hyperparameters for the six datasets are listed in Table 6.

**Table 4:** Comparison of clustering performance on the benchmark datasets under different missing-view rates (MR) of 10%, 30%, 50%, 70%. The best results are highlighted in **bold**, while the second-best are underlined. Each experiment is conducted five times, with the mean and standard deviation reported.

| MR | 10% | | | | 30% | | | | 50% | | | | 70% | | | |
|---|---|---|---|---|---|---|---|---|---|---|---|---|---|---|---|---|
| Metrics | ACC↑ | NMI↑ | ARI↑ | PUR↑ | ACC↑ | NMI↑ | ARI↑ | PUR↑ | ACC↑ | NMI↑ | ARI↑ | PUR↑ | ACC↑ | NMI↑ | ARI↑ | PUR↑ |
| **Scene15** | | | | | | | | | | | | | | | | |
| BSV | 0.3595±0.0061 | 0.3577±0.0054 | 0.1784±0.0041 | 0.4002±0.0092 | 0.3305±0.0086 | 0.3206±0.0071 | 0.1253±0.0051 | 0.3630±0.0097 | 0.2944±0.0100 | 0.2905±0.0063 | 0.0796±0.0046 | 0.3321±0.0132 | 0.2593±0.0085 | 0.2568±0.0030 | 0.0472±0.0015 | 0.2903±0.0029 |
| Concat | 0.3703±0.0038 | 0.3820±0.0024 | 0.2037±0.0031 | 0.4263±0.0031 | 0.3360±0.0067 | 0.3318±0.0035 | 0.1540±0.0059 | 0.3825±0.0059 | 0.2943±0.0139 | 0.2995±0.0083 | 0.1201±0.0057 | 0.3436±0.0089 | 0.2596±0.0137 | 0.2665±0.0020 | 0.0982±0.0042 | 0.3053±0.0084 |
| DCP | 0.4071±0.0109 | 0.4414±0.0098 | 0.2488±0.0158 | 0.4342±0.0158 | 0.3958±0.0072 | 0.4264±0.0052 | 0.2350±0.0084 | 0.4265±0.0084 | 0.3879±0.0216 | 0.4095±0.0101 | 0.2359±0.0063 | 0.4166±0.0161 | 0.3784±0.0084 | 0.3849±0.0125 | 0.2113±0.0113 | 0.4026±0.0100 |
| CPSPAN | 0.3574±0.0201 | 0.3457±0.0148 | 0.1911±0.0142 | 0.4103±0.0188 | 0.3161±0.0137 | 0.2783±0.0118 | 0.1541±0.0076 | 0.3439±0.0173 | 0.2553±0.0071 | 0.2070±0.0076 | 0.1063±0.0049 | 0.2786±0.0058 | 0.2025±0.0117 | 0.1423±0.0138 | 0.0621±0.0101 | 0.2198±0.0126 |
| DSIMVC | 0.2295±0.0142 | 0.2976±0.0107 | 0.1430±0.0063 | 0.3218±0.0101 | 0.2734±0.0101 | 0.2896±0.0087 | 0.1383±0.0067 | 0.3131±0.0058 | 0.2697±0.0059 | 0.2828±0.0069 | 0.1335±0.0043 | 0.3133±0.0069 | 0.2654±0.0090 | 0.2741±0.0082 | 0.1290±0.0067 | 0.3062±0.0079 |
| DVIMC | 0.4863±0.0103 | 0.4780±0.0059 | 0.3192±0.0085 | 0.5021±0.0109 | 0.4678±0.0084 | 0.4440±0.0033 | 0.3097±0.0033 | 0.4823±0.0089 | 0.4371±0.0568 | 0.4163±0.0402 | 0.2737±0.0399 | 0.4577±0.0654 | 0.4136±0.0266 | 0.3950±0.0228 | 0.2519±0.0189 | 0.4286±0.0293 |
| MVP | 0.4451±0.0188 | 0.4361±0.0112 | 0.2696±0.0126 | 0.4830±0.0161 | 0.4458±0.0064 | 0.4341±0.0061 | 0.2723±0.0076 | 0.4866±0.0099 | 0.4564±0.0068 | 0.4314±0.0074 | 0.2967±0.0095 | 0.5170±0.0130 | 0.4407±0.0163 | 0.4285±0.0091 | 0.2725±0.0117 | 0.4895±0.0171 |
| PMIMC | 0.3317±0.0032 | 0.3409±0.0053 | 0.1778±0.0032 | 0.3602±0.0073 | 0.3071±0.0118 | 0.3175±0.0028 | 0.1484±0.0036 | 0.3500±0.0086 | 0.3125±0.0060 | 0.3180±0.0014 | 0.1477±0.0026 | 0.3513±0.0061 | 0.3178±0.0071 | 0.3173±0.0041 | 0.1492±0.0036 | 0.3509±0.0064 |
| ACOVA (Ours) | **0.5040±0.0153** | **0.4804±0.0050** | **0.3273±0.0088** | **0.5270±0.0202** | **0.4807±0.0136** | **0.4560±0.0090** | **0.3106±0.0090** | **0.5119±0.0164** | **0.4710±0.0138** | **0.4360±0.0034** | **0.2973±0.0084** | 0.4987±0.0126 | **0.4428±0.0140** | 0.4083±0.0055 | **0.2733±0.0031** | 0.4636±0.0161 |
| **CaltechV5-7** | | | | | | | | | | | | | | | | |
| BSV | 0.5678±0.0051 | 0.4364±0.0045 | 0.3408±0.0046 | 0.5827±0.0050 | 0.5035±0.0086 | 0.3699±0.0049 | 0.2308±0.0032 | 0.5170±0.0064 | 0.4483±0.0062 | 0.3235±0.0046 | 0.1488±0.0027 | 0.4617±0.0053 | 0.3859±0.0043 | 0.2798±0.0055 | 0.0890±0.0032 | 0.4019±0.0062 |
| Concat | 0.4750±0.0337 | 0.3300±0.0136 | 0.2519±0.0232 | 0.4957±0.0186 | 0.4450±0.0128 | 0.2941±0.0076 | 0.1998±0.0077 | 0.4593±0.0091 | 0.4279±0.0140 | 0.2756±0.0133 | 0.1595±0.0081 | 0.4386±0.0132 | 0.3600±0.0241 | 0.2338±0.0201 | 0.0891±0.0110 | 0.3757±0.0172 |
| DCP | 0.5754±0.0185 | 0.5719±0.0175 | 0.4612±0.0339 | 0.6054±0.0177 | 0.5363±0.0206 | 0.5269±0.0132 | 0.4199±0.0212 | 0.5751±0.0113 | 0.5063±0.0209 | 0.4729±0.0255 | 0.3406±0.0454 | 0.5393±0.0289 | 0.3982±0.0289 | 0.3363±0.0156 | 0.1378±0.0234 | 0.4056±0.0276 |
| CPSPAN | 0.7689±0.0185 | 0.6677±0.0221 | 0.6080±0.0298 | 0.7750±0.0176 | 0.6874±0.0359 | 0.5653±0.0265 | 0.4666±0.0310 | 0.6688±0.0343 | 0.6010±0.0249 | 0.4789±0.0343 | 0.3467±0.0184 | 0.6030±0.0249 | 0.4861±0.0389 | 0.3464±0.0329 | 0.2222±0.0309 | 0.4904±0.0373 |
| DSIMVC | 0.7760±0.0536 | 0.6998±0.0245 | 0.6280±0.0395 | 0.7816±0.0440 | 0.7841±0.0261 | 0.6947±0.0206 | 0.6335±0.0289 | 0.7841±0.0261 | 0.7339±0.0370 | 0.6868±0.0088 | 0.5989±0.0172 | 0.7413±0.0307 | 0.7029±0.0193 | 0.5838±0.0212 | 0.5083±0.0227 | 0.7047±0.0162 |
| DVIMC | 0.9013±0.0103 | 0.8207±0.0137 | 0.7999±0.0182 | 0.9013±0.0103 | 0.8921±0.0093 | 0.8011±0.0166 | 0.7850±0.0170 | 0.8921±0.0093 | 0.8609±0.0123 | 0.7507±0.0186 | 0.7322±0.0186 | 0.8609±0.0123 | 0.8514±0.0050 | 0.7345±0.0103 | 0.7195±0.0080 | 0.8514±0.0050 |
| MVP | 0.8083±0.0148 | 0.6969±0.0212 | 0.6594±0.0242 | 0.8083±0.0148 | 0.8070±0.0213 | 0.7139±0.0172 | 0.6723±0.0208 | 0.8146±0.0132 | 0.8103±0.0231 | 0.7063±0.0235 | 0.6700±0.0283 | 0.8157±0.0184 | 0.7907±0.0354 | 0.6483±0.0333 | 0.5934±0.0432 | 0.7693±0.0337 |
| PMIMC | 0.8963±0.0120 | 0.8205±0.0171 | 0.7966±0.0251 | 0.8963±0.0120 | 0.8917±0.0118 | 0.8124±0.0117 | 0.7858±0.0251 | 0.8917±0.0128 | 0.8601±0.0071 | 0.7591±0.0312 | 0.7381±0.0141 | 0.8601±0.0071 | 0.8570±0.0119 | 0.7522±0.0092 | 0.7304±0.0141 | 0.8570±0.0119 |
| ACOVA (Ours) | **0.9149±0.0129** | **0.8507±0.0190** | **0.8287±0.0237** | **0.9149±0.0129** | **0.9131±0.0186** | **0.8435±0.0233** | **0.8226±0.0334** | **0.9131±0.0186** | **0.8953±0.0151** | **0.8078±0.0230** | **0.7918±0.0270** | **0.8953±0.0151** | **0.8660±0.0075** | **0.7531±0.0146** | **0.7340±0.0146** | **0.8660±0.0075** |
| **Handwritten** | | | | | | | | | | | | | | | | |
| BSV | 0.7087±0.0113 | 0.6647±0.0036 | 0.5625±0.0083 | 0.7433±0.0022 | 0.7105±0.0435 | 0.6344±0.0255 | 0.4716±0.0336 | 0.7171±0.0368 | 0.5842±0.0145 | 0.5444±0.0145 | 0.3028±0.0193 | 0.5970±0.0182 | 0.5159±0.0286 | 0.4802±0.0197 | 0.1874±0.0175 | 0.5208±0.0273 |
| Concat | 0.7684±0.0283 | 0.7214±0.0112 | 0.6412±0.0149 | 0.7837±0.0174 | 0.7147±0.0202 | 0.6441±0.0094 | 0.5012±0.0131 | 0.7190±0.0120 | 0.6108±0.0596 | 0.5437±0.0466 | 0.3492±0.0172 | 0.6166±0.0524 | 0.5021±0.0203 | 0.4498±0.0129 | 0.2511±0.0117 | 0.5105±0.0157 |
| DCP | 0.6459±0.0434 | 0.7094±0.0154 | 0.5663±0.0251 | 0.6737±0.0296 | 0.6639±0.0462 | 0.6951±0.0270 | 0.5189±0.0638 | 0.6877±0.0357 | 0.7045±0.0411 | 0.7018±0.0218 | 0.5651±0.0397 | 0.7334±0.0354 | 0.6470±0.0349 | 0.6398±0.0287 | 0.3289±0.0251 | 0.6486±0.0335 |
| CPSPAN | 0.8710±0.0485 | 0.7969±0.0267 | 0.7586±0.0462 | 0.8727±0.0452 | 0.7524±0.0423 | 0.6927±0.0183 | 0.5603±0.0301 | 0.7638±0.0327 | 0.6430±0.0415 | 0.6050±0.0145 | 0.3862±0.0130 | 0.6595±0.0331 | 0.6913±0.0457 | 0.5828±0.0219 | 0.4856±0.0328 | 0.7083±0.0352 |
| DSIMVC | 0.8014±0.0429 | 0.7960±0.0355 | 0.7282±0.0471 | 0.8158±0.0301 | 0.8180±0.0466 | 0.8082±0.0427 | 0.7455±0.0570 | 0.8275±0.0352 | 0.8304±0.0327 | 0.8122±0.0256 | 0.7543±0.0358 | 0.8342±0.0252 | 0.8284±0.0093 | 0.7879±0.0110 | 0.7312±0.0124 | 0.8284±0.0093 |
| DVIMC | 0.9169±0.0545 | 0.8604±0.0268 | 0.8284±0.0444 | 0.9201±0.0481 | 0.8954±0.0515 | 0.8470±0.0336 | 0.8210±0.0527 | 0.8920±0.0486 | 0.8980±0.0486 | 0.8373±0.0199 | 0.8175±0.0397 | 0.9054±0.0340 | 0.8099±0.0403 | 0.7808±0.0560 | 0.7197±0.0412 | 0.8099±0.0403 |
| MVP | 0.7883±0.0402 | 0.8002±0.0174 | 0.7114±0.0256 | 0.8119±0.0297 | 0.8095±0.0169 | 0.8000±0.0267 | 0.7270±0.0265 | 0.8208±0.0117 | 0.7474±0.0597 | 0.7853±0.0111 | 0.6834±0.0371 | 0.7715±0.0408 | 0.7483±0.0316 | 0.8325±0.0115 | 0.7758±0.0242 | 0.7678±0.0309 |
| PMIMC | 0.8826±0.0367 | 0.8188±0.0132 | 0.7846±0.0266 | 0.8834±0.0352 | 0.8998±0.0115 | 0.8213±0.0319 | 0.7962±0.0194 | 0.8998±0.0115 | 0.8613±0.0557 | 0.8096±0.0254 | 0.7637±0.0519 | 0.8695±0.0427 | 0.8522±0.0588 | 0.7994±0.0295 | 0.7505±0.0574 | 0.8587±0.0503 |
| ACOVA (Ours) | **0.9363±0.0067** | **0.8692±0.0113** | **0.8645±0.0133** | **0.9363±0.0067** | **0.9376±0.0073** | **0.8709±0.0125** | **0.8669±0.0144** | **0.9376±0.0073** | **0.9146±0.0321** | **0.8489±0.0280** | **0.8325±0.0453** | **0.9146±0.0321** | **0.8999±0.0526** | **0.8351±0.0325** | **0.8176±0.0541** | **0.9021±0.0483** |
| **Fashion** | | | | | | | | | | | | | | | | |
| BSV | 0.4832±0.0163 | 0.4606±0.0100 | 0.2986±0.0076 | 0.5210±0.0113 | 0.4424±0.0167 | 0.4114±0.0080 | 0.2210±0.0048 | 0.4779±0.0360 | 0.4100±0.0245 | 0.3734±0.0074 | 0.1626±0.0050 | 0.4361±0.0121 | 0.3645±0.0144 | 0.3348±0.0025 | 0.1040±0.0011 | 0.3916±0.0085 |
| Concat | 0.6546±0.0575 | 0.6518±0.0254 | 0.5451±0.0369 | 0.7028±0.0499 | 0.5002±0.0302 | 0.4618±0.0223 | 0.3646±0.0198 | 0.5272±0.0214 | 0.4146±0.0058 | 0.3433±0.0120 | 0.2509±0.0114 | 0.4350±0.0089 | 0.2988±0.0335 | 0.1976±0.0200 | 0.1307±0.0098 | 0.3247±0.0250 |
| DCP | 0.8547±0.0618 | 0.8864±0.0188 | 0.8119±0.0519 | 0.8652±0.0547 | 0.7663±0.0494 | 0.8340±0.0177 | 0.7285±0.0371 | 0.7899±0.0473 | 0.7533±0.0497 | 0.7995±0.0166 | 0.6904±0.0356 | 0.7643±0.0461 | 0.6717±0.0163 | 0.7430±0.0077 | 0.6079±0.0154 | 0.6879±0.0154 |
| CPSPAN | 0.5002±0.0302 | 0.5451±0.0369 | 0.5451±0.0369 | 0.7028±0.0540 | 0.5002±0.0302 | 0.4618±0.0223 | 0.3646±0.0198 | 0.5272±0.0214 | 0.4146±0.0058 | 0.3433±0.0120 | 0.2509±0.0114 | 0.4350±0.0089 | 0.2988±0.0335 | 0.1976±0.0200 | 0.1307±0.0098 | 0.3247±0.0250 |
| DSIMVC | 0.8993±0.0236 | 0.8573±0.0152 | 0.8178±0.0317 | 0.8993±0.0236 | 0.8810±0.0298 | 0.8772±0.0172 | 0.8162±0.0374 | 0.8810±0.0298 | 0.8271±0.0256 | 0.7990±0.0117 | 0.7402±0.0236 | 0.7403±0.0256 | 0.8064±0.0256 | 0.7707±0.0117 | 0.6951±0.0236 | 0.7904±0.0256 |
| DVIMC | 0.8856±0.0541 | 0.8799±0.0126 | 0.8284±0.0444 | 0.9024±0.0337 | 0.8806±0.0461 | 0.8718±0.0098 | 0.8236±0.0372 | 0.8996±0.0301 | 0.8629±0.0627 | 0.8449±0.0217 | 0.7965±0.0482 | 0.8867±0.0505 | 0.8579±0.0485 | 0.8290±0.0102 | 0.7729±0.0406 | 0.8725±0.0310 |
| MVP | 0.8691±0.0550 | 0.8743±0.0175 | 0.8175±0.0473 | 0.8955±0.0339 | 0.8259±0.0025 | 0.8537±0.0021 | 0.7744±0.0029 | 0.8722±0.0016 | 0.7982±0.0441 | 0.8372±0.0155 | 0.7484±0.0338 | 0.8485±0.0384 | 0.8386±0.0619 | 0.8325±0.0191 | 0.7758±0.0493 | 0.8665±0.0407 |
| PMIMC | 0.6986±0.0797 | 0.7563±0.0237 | 0.6282±0.0535 | 0.7427±0.0540 | 0.7039±0.0319 | 0.6593±0.0250 | 0.6365±0.0268 | 0.7535±0.0308 | 0.7118±0.0368 | 0.7586±0.0208 | 0.6295±0.0321 | 0.7577±0.0373 | 0.7059±0.0334 | 0.7583±0.0154 | 0.6362±0.0262 | 0.7540±0.0318 |
| ACOVA (Ours) | **0.9056±0.0627** | **0.8876±0.0208** | **0.8516±0.0494** | **0.9129±0.0481** | **0.8848±0.0562** | **0.8735±0.0141** | **0.8266±0.0460** | **0.9009±0.0365** | **0.8764±0.0375** | **0.8461±0.0150** | **0.7815±0.0456** | **0.8736±0.0343** | **0.8503±0.0536** | **0.8461±0.0150** | **0.7815±0.0456** | **0.8736±0.0343** |
| **NoisyMNIST** | | | | | | | | | | | | | | | | |
| BSV | 0.5144±0.0037 | 0.4542±0.0087 | 0.3258±0.0035 | 0.5618±0.0029 | 0.4184±0.0645 | 0.3343±0.0899 | 0.1896±0.0604 | 0.4456±0.0837 | 0.2939±0.0051 | 0.1912±0.0024 | 0.0754±0.0022 | 0.3090±0.0043 | 0.2672±0.0048 | 0.1566±0.0031 | 0.0452±0.0007 | 0.2693±0.0056 |
| Concat | 0.4338±0.0016 | 0.3910±0.0012 | 0.2727±0.0009 | 0.4418±0.0009 | 0.3873±0.0057 | 0.3229±0.0065 | 0.1891±0.0047 | 0.4036±0.0022 | 0.3463±0.0210 | 0.2592±0.0245 | 0.1252±0.0278 | 0.3561±0.0187 | 0.2805±0.0248 | 0.1832±0.0278 | 0.0615±0.0299 | 0.2864±0.0216 |
| DCP | 0.8945±0.0063 | 0.9149±0.0355 | 0.8607±0.0139 | 0.9319±0.0439 | 0.8997±0.0755 | 0.8952±0.0845 | 0.8608±0.0845 | 0.9198±0.0522 | 0.9150±0.0286 | 0.8586±0.0222 | 0.8401±0.0499 | 0.9150±0.0286 | 0.9225±0.0185 | 0.8335±0.0127 | 0.8431±0.0274 | 0.9225±0.0185 |
| CPSPAN | 0.5665±0.0149 | 0.5870±0.0250 | 0.4402±0.0277 | 0.6058±0.0149 | 0.6103±0.0311 | 0.6050±0.0193 | 0.4724±0.0295 | 0.6357±0.0180 | 0.5774±0.0170 | 0.5824±0.0231 | 0.4439±0.0201 | 0.6097±0.0166 | 0.5486±0.0166 | 0.5766±0.0185 | 0.4271±0.0291 | 0.5932±0.0213 |
| DSIMVC | 0.8993±0.0236 | 0.8573±0.0152 | 0.8178±0.0317 | 0.8977±0.1654 | 0.7337±0.1270 | 0.6790±0.1017 | 0.6148±0.1417 | 0.7464±0.1211 | 0.6152±0.1458 | 0.5769±0.1145 | 0.4848±0.1589 | 0.6361±0.1331 | 0.6245±0.0276 | 0.5822±0.0283 | 0.4867±0.0321 | 0.6343±0.0347 |
| DVIMC | 0.9372±0.0361 | 0.9345±0.0174 | 0.9157±0.0533 | 0.9466±0.0423 | 0.9365±0.0395 | 0.9060±0.0174 | 0.8987±0.0409 | 0.9368±0.0390 | 0.8898±0.0479 | 0.8699±0.0232 | 0.8558±0.0469 | 0.9019±0.0460 | 0.8741±0.0562 | 0.8184±0.0292 | 0.7999±0.0560 | 0.8763±0.0520 |
| MVP | 0.8657±0.0344 | 0.8778±0.0383 | 0.8176±0.0571 | 0.8667±0.0428 | 0.8590±0.1030 | 0.8635±0.0521 | 0.8222±0.1028 | 0.8777±0.0821 | 0.8355±0.0337 | 0.8195±0.0318 | 0.7792±0.0407 | 0.8464±0.0321 | 0.6660±0.0712 | 0.6744±0.1043 | 0.5721±0.1088 | 0.6694±0.0713 |
| PMIMC | 0.7127±0.0953 | 0.6985±0.0616 | 0.5927±0.0961 | 0.7432±0.0848 | 0.7187±0.0147 | 0.6593±0.0250 | 0.5623±0.0119 | 0.7458±0.0163 | 0.6616±0.0362 | 0.5739±0.0194 | 0.4947±0.0240 | 0.6695±0.0301 | 0.6044±0.0243 | 0.5544±0.0259 | 0.4393±0.0259 | 0.6220±0.0233 |
| ACOVA (Ours) | **0.9663±0.0393** | **0.9478±0.0240** | **0.9440±0.0508** | **0.9664±0.0393** | **0.9344±0.0471** | **0.9125±0.0195** | **0.9016±0.0487** | **0.9348±0.0465** | **0.9599±0.0040** | **0.9001±0.0023** | **0.9144±0.0020** | **0.9599±0.0010** | **0.9377±0.0080** | **0.8579±0.0111** | **0.8697±0.0152** | **0.9377±0.0080** |
| **CUB Image-Caption** | | | | | | | | | | | | | | | | |
| BSV | 0.6723±0.0381 | 0.6604±0.0149 | 0.4818±0.0237 | 0.6823±0.0247 | 0.6080±0.0294 | 0.5858±0.0118 | 0.3376±0.0252 | 0.6163±0.0208 | 0.5270±0.0188 | 0.5241±0.0062 | 0.2295±0.0102 | 0.5387±0.0193 | 0.4493±0.0407 | 0.4474±0.0243 | 0.1266±0.0166 | 0.4630±0.0335 |
| CONCAT | 0.6890±0.0291 | 0.6685±0.0131 | 0.4979±0.0228 | 0.6947±0.0212 | 0.5837±0.0370 | 0.5790±0.0229 | 0.3322±0.0400 | 0.5987±0.0269 | 0.5187±0.0295 | 0.5107±0.0101 | 0.2068±0.0202 | 0.5323±0.0157 | 0.4663±0.0214 | 0.4658±0.0146 | 0.1391±0.0157 | 0.4770±0.0234 |
| DSIMVC | 0.5227±0.0296 | 0.5822±0.0087 | 0.2867±0.0139 | 0.5833±0.0118 | 0.5163±0.0580 | 0.5781±0.0371 | 0.3008±0.0559 | 0.5797±0.0359 | 0.4040±0.0235 | 0.4430±0.0186 | 0.1196±0.0180 | 0.4587±0.0202 | 0.4830±0.0085 | 0.2337±0.0201 | 0.0571±0.0080 | 0.2867±0.0166 |
| DCP | 0.6687±0.0096 | 0.6764±0.0079 | 0.6356±0.0112 | 0.5396±0.0129 | 0.5560±0.0347 | 0.5603±0.0179 | 0.5767±0.0359 | 0.3731±0.0240 | 0.5497±0.0100 | 0.5349±0.0142 | 0.5100±0.0176 | 0.2823±0.0188 | 0.4810±0.0177 | 0.4810±0.0177 | 0.4547±0.0126 | 0.1933±0.0432 |
| CSPAN | 0.6380±0.0188 | 0.5744±0.0157 | 0.4288±0.0159 | 0.6427±0.0146 | 0.6420±0.0105 | 0.5663±0.0088 | 0.4236±0.0102 | 0.6420±0.0105 | 0.5810±0.0257 | 0.5419±0.0287 | 0.3824±0.0319 | 0.5860±0.0224 | 0.4965±0.0419 | 0.4965±0.0419 | 0.3275±0.0544 | 0.5440±0.0541 |
| DVIMC | 0.6573±0.0427 | 0.6585±0.0266 | 0.5246±0.0366 | 0.6807±0.0383 | 0.6703±0.0372 | 0.6502±0.0409 | 0.5131±0.0448 | 0.6807±0.0334 | 0.6057±0.0341 | 0.6026±0.0252 | 0.4538±0.0285 | 0.6160±0.0233 | 0.4710±0.0438 | 0.4694±0.0452 | 0.2795±0.0477 | 0.4803±0.0489 |
| MVP | 0.6497±0.0246 | 0.6317±0.0154 | 0.5051±0.0139 | 0.6667±0.0176 | 0.6840±0.0538 | 0.6545±0.0263 | 0.5256±0.0315 | 0.6903±0.0325 | 0.6163±0.0315 | 0.5750±0.0178 | 0.4401±0.0250 | 0.6287±0.0373 | 0.6440±0.0274 | 0.6036±0.0198 | 0.4655±0.0221 | 0.6510±0.0211 |
| PMIMC | 0.7183±0.0953 | 0.7131±0.0108 | 0.5795±0.0072 | 0.7383±0.0053 | 0.6820±0.0491 | 0.6847±0.0221 | 0.5360±0.0412 | 0.6847±0.0221 | 0.6517±0.0290 | 0.6198±0.0163 | 0.4846±0.0228 | 0.6590±0.0253 | 0.5907±0.0515 | 0.5480±0.0315 | 0.4034±0.0367 | 0.6060±0.0475 |
| ACOVA | **0.7970±0.0143** | **0.7695±0.0139** | **0.6581±0.0195** | **0.7970±0.0143** | **0.7563±0.0242** | **0.7335±0.0085** | **0.6215±0.0242** | **0.7590±0.0216** | **0.6823±0.0049** | **0.6568±0.0193** | **0.5267±0.0235** | **0.6917±0.0252** | **0.6554±0.0310** | **0.6136±0.0223** | **0.4730±0.0216** | **0.6584±0.0229** |

**Table 5:** Statistics and brief summary of eight comparison methods.

| Method | Venue | #View | Description |
|---|---|---|---|
| BSV (Zhao et al., 2016) | IJCAI | Multiple | Imputes missing views using the view-wise average values and performs clustering using the best-performing single view. |
| CONCAT (Zhao et al., 2016) | IJCAI | Multiple | Similar to BSV that imputes missing views using the view-wise average values, but concatenates features across all views after imputation and then performs $K$-Means clustering. |
| DSIMVC (Tang & Liu, 2022) | ICML | Multiple | A bi-level optimization-based safe incomplete multi-view clustering framework that dynamically imputes missing views from semantic neighbors and automatically selects reliable imputed samples. |
| DCP (Lin et al., 2023) | TPAMI | Two | An information-theoretical framework that unifies cross-view consistency learning and missing-view recovery via dual contrastive learning and dual prediction. |
| CPSPAN (Jin et al., 2023) | CVPR | Multiple | Jointly performs cross-view partial sample alignment and prototype-level alignment to address the challenges of prototype shift under missing-view settings. |
| DVIMC (Xu et al., 2024) | AAAI | Multiple | Employ PoE-based VAEs to construct a shared latent space and mitigate information imbalance via a coherence loss. |
| MVP (Gao & Pu, 2025) | ICLR | Multiple | A variational framework that explicitly infers missing views in latent space and enhances consistency via cyclic permutation-based regularization. |
| PMIMC (Yuan et al., 2025) | TIP | Multiple | An IMVC method that solves the prototype-unaligned problem and performance instability problem by prototype contrastive learning loss and prototype-based imputation strategies. |

**Our source code is attached in the supplementary file.**

**Table 6:** Hyperparameter settings for all datasets. "ExpLR" is short for ExponentialLR.

| Hyperparameters | Handwritten | Caltech5V | Scene15 | Fashion-MV | NosiyMNIST | CUB |
|---|---|---|---|---|---|---|
| Pretraining Epochs | 200 | 200 | 200 | 200 | 200 | 200 |
| Training Epochs | 300 | 300 | 300 | 300 | 300 | 300 |
| Batch Size | 256 | 256 | 256 | 256 | 512 | 256 |
| Optimizer | Adam | Adam | Adam | Adam | Adam | Adam |
| Scheduler | ExpLR | ExpLR | ExpLR | ExpLR | ExpLR | ExpLR |
| Network Learning Rate | $3e^{-4}$ | $1e^{-3}$ | $1e^{-3}$ | $1e^{-3}$ | $1e^{-3}$ | $5e^{-4}$ |
| Prior Distribution Learning Rate | $1e^{-2}$ | $1e^{-1}$ | $5e^{-2}$ | $5e^{-2}$ | $5e^{-2}$ | $5e^{-2}$ |
| Correlation Learning Rate | $1e^{-2}$ | $1e^{-2}$ | $1e^{-2}$ | $1e^{-2}$ | $1e^{-2}$ | $1e^{-2}$ |
| Balance parameter $\alpha$ | 15 | 5 | 20 | 20 | 10 | 60 |
| Latent feature dimension $D$ | 10 | 15 | 10 | 10 | 10 | 10 |

# C  ADDITIONAL EXPERIMENTAL RESULTS

## C.1  COMPLETE RESULTS ON THE SIX DATASETS

In Table 4, we report the complete results, including the mean and standard deviation over five runs for all methods across the six datasets.

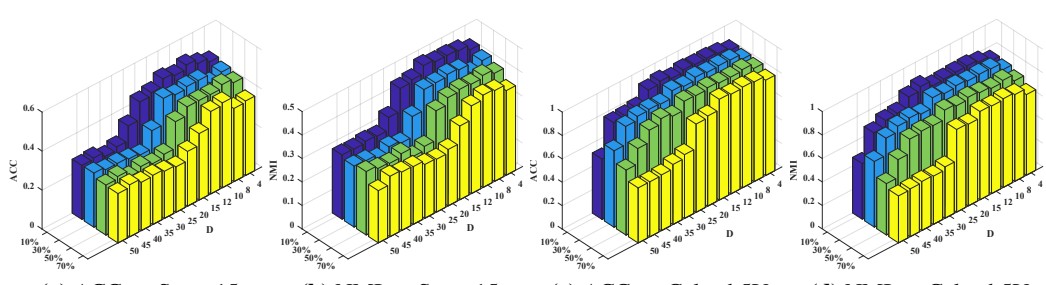

**(a)** ACC on Scene15     **(b)** NMI on Scene15     **(c)** ACC on Caltech5V     **(d)** NMI on Caltech5V

**Figure 6:** Clustering performance with different latent dimension $D$ and missing rate settings on Scene15 and Caltech5V dataset.

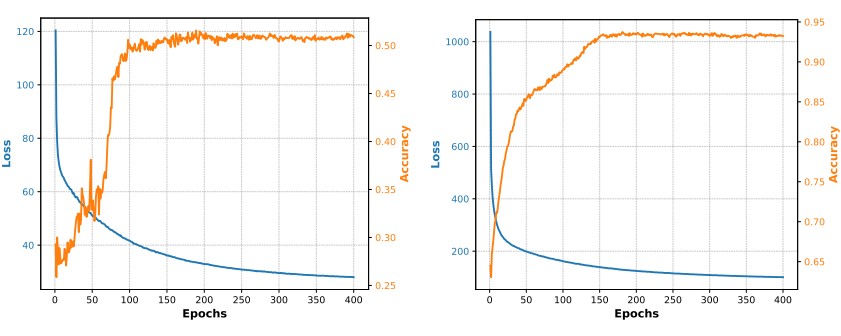

**(a)** Scene15        **(b)** Handwritten

**Figure 7:** Convergence curves on Scene15 and Handwritten dataset.

## C.2   IMPACT OF LATENT DIMENSION $D$

As illustrated in Figure 6, we empirically investigate the impact of latent dimension $D$ on clustering performance across different missing rates on Scene15 and Caltech5V datasets. We examine $D$ ranging from {4, 8, 10, 12, 15, 20, 25, 30, 35, 40, 45, 50 }, and observe that increasing $D$ from 4 to 15 leads to performance improvements, while further increases yield metric degradation. This trend remains consistent across different missing ratios and datasets. Empirically, we recommend setting $D$ in the range of 10-15 for optimal clustering performance.

## C.3   TIME COST ANALYSIS

In Table 7 we report the training time and clustering accuracy (ACC) of all compared methods on the Scene15 dataset with a 70% missing ratio. Note that BSV and CONCAT do not require any training procedures, and thus we omit their time cost in the table. Furthermore, for fair comparison, we report the training time for the main algorithm and exclude the pretraining stage. The results demonstrate that our ACOVA achieves an effective trade-off between computational cost and clustering performance, exhibiting moderate training time while maintaining superior clustering accuracy. We further repeat the analysis on the NoisyMNIST dataset, due to its larger scale, demonstrating explicitly the beneficial scaling of our approach when it comes to the number of samples $N$. To show the impact of $V$, we also conducted the runtime analysis for different numbers of views on the Caltech5V dataset in Table 8. It can be observed that the overhead of the cubic scaling with respect to $V$ is negligible for most common settings.

**Table 7:** Runtime comparison on the Scene15 and NoisyMNIST datasets.

| Dataset | Metric | BSV | CONCAT | DSIMVC | DCP | CSPAN | DVIMC | MVP | PMIMC | ACOVA |
|---------|--------|-----|--------|--------|-----|-------|-------|-----|-------|-------|
| Scene15 | Time | - | - | $2.35 \times 10^{-2}$ | $1.47 \times 10^{1}$ | $6.62 \times 10^{0}$ | $1.12 \times 10^{-1}$ | $1.84 \times 10^{0}$ | $4.63 \times 10^{0}$ | $1.16 \times 10^{-1}$ |
|  | ACC | 0.2197 | 0.2497 | 0.3130 | 0.3699 | 0.3199 | 0.3458 | 0.4130 | 0.3093 | 0.4546 |
| NoisyMNIST | Time | - | - | $3.06 \times 10^{0}$ | $1.48 \times 10^{1}$ | $2.22 \times 10^{2}$ | $4.98 \times 10^{0}$ | $7.61 \times 10^{1}$ | $1.14 \times 10^{2}$ | $5.59 \times 10^{0}$ |
|  | ACC | 0.2672 | 0.2805 | 0.6245 | 0.9225 | 0.5486 | 0.8741 | 0.6660 | 0.6044 | 0.9377 |

**Table 8:** Runtime analysis with varying number of views on Caltech5V dataset

| Number of Views ($V$) | 2 | 3 | 4 | 5 |
|---|---|---|---|---|
| Runtime (seconds) | 0.18 ± 0.03 | 0.22 ± 0.03 | 0.26 ± 0.04 | 0.27 ± 0.04 |

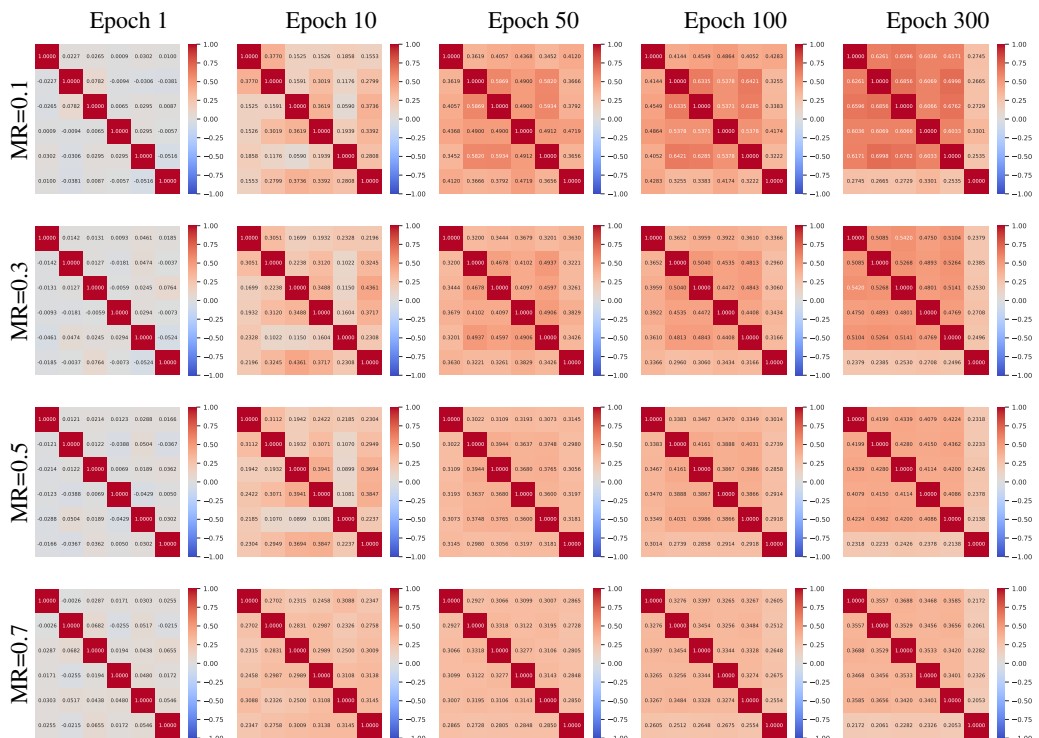

**(a)** Caltech5V            **(b)** Handwritten

**Figure 8:** Convergence curves of $\|\mathbf{R} - \mathbf{I}\|_F$ on Caltech5V and Handwritten dataset.

## C.4 CONVERGENCE ANALYSIS

To verify the convergence, we analyze both the training objective (Loss) and clustering performance (Accuracy) against epochs on the Scene15 and Handwritten datasets with a 10% missing ratio. As illustrated in Figure 7, the training objective exhibits consistent monotonic descent until convergence, while the clustering accuracy steadily increases and eventually saturates. These results all confirm good convergence properties of the proposed ACOVA.

## C.5 ANALYSIS OF LEARNED CORRELATION MATRIX

**Figure 9:** Correlation matrices on Handwritten: rows are missing rates, columns are epochs.

In this section, we further analyze the process of learning the correlation matrix. In Figure 8, we monitor $\|\mathbf{R} - \mathbf{I}\|_F$ throughout training on the Caltech5V and Handwritten datasets. From these results, we observe that $\|\mathbf{R} - \mathbf{I}\|_F$ converges to values between 1.2 and 2.9 for the Caltech5V dataset and values between 1.4 and 3 for the Handwritten dataset. This lies well within the theoretical range of $[0, \sqrt{20}]$ and $[0, \sqrt{30}]$, respectively. We further observe that for large missingness, the correlation

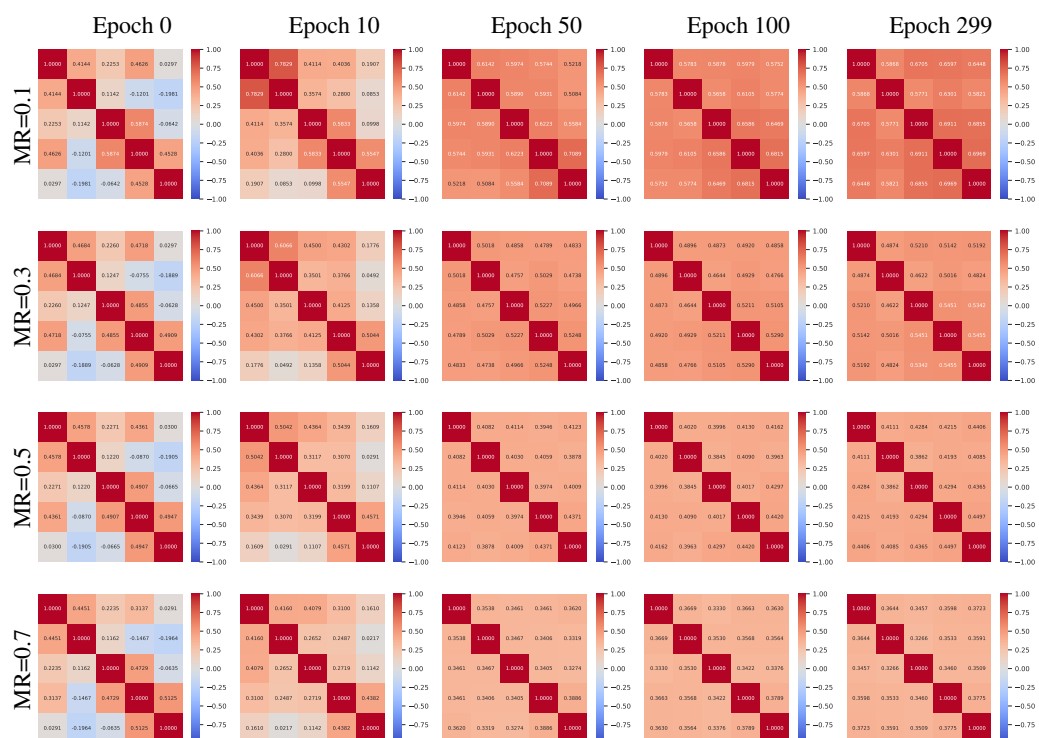

**Figure 10:** Correlation matrices on Caltech5V: rows are missing rates, columns are epochs.

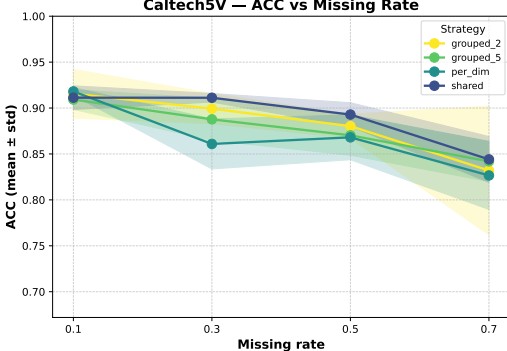

**Figure 11:** Clustering accuracy (ACC) of different strategies on the Caltech5V dataset under varying missing rates. The proposed shared correlation matrix achieves consistently strong performance compared with other strategies.

matrix is closer to the identity, while for low missingness the correlation between views is higher. This intuitively makes sense as less samples are being observed with paired views.

Figure 9 and Figure 10 further demonstrate the evolution of $\mathbf{R}$ throughout training. For the Caltech5V dataset, we observe the strongest correlation between the HOG and LBP features, which both capture local texture and edge information. This further validates that our approach learns a meaningful correlation structure.

Finally, we further empirically demonstrate that $\mathbf{R}$ is stable across runs. For this we conduct five runs with different random seeds on the Handwritten dataset and compute the variance across runs for the different elements in $\mathbf{R}$. Figure 12 provides these results for 10% and 90% missingess, demonstrating high stability even for large missing-rates.

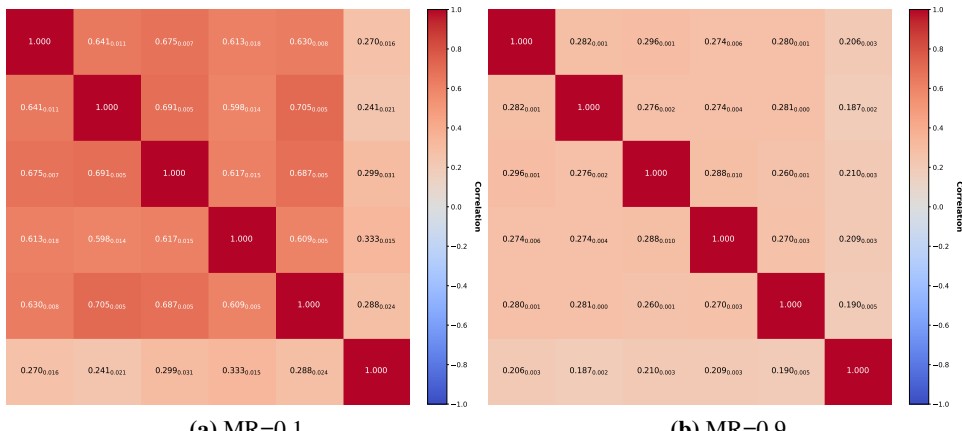

**(a)** MR=0.1          **(b)** MR=0.9

**Figure 12:** Correlation matrices for the Handwritten dataset averaged across five runs with mean and standard deviation.

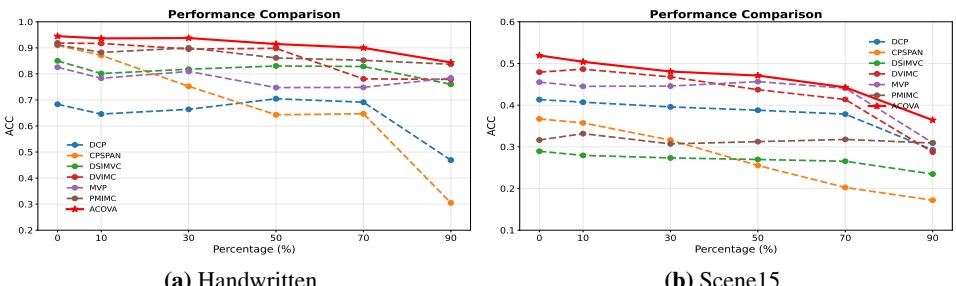

**(a)** Handwritten          **(b)** Scene15

**Figure 13:** Performance curves vs. missing rates. The proposed approach consistently outperforms prior approaches even in the extreme scenarios.

## C.6 LEARNING MULTIPLE CORRELATION MATRICES

As our proposed framework enables end-to-end optimization of $\mathbf{R}$, we can further extend our framework to learn multiple correlation matrices. In Fig. 11, we compare the performance of learning one shared $\mathbf{R}$ to learning a separate $\mathbf{R}$ for each dimension or for groups of dimensions. More specifically, "grouped_2" and "grouped_5" corresponds to learning two and five $\mathbf{R}$ matrices for groups of $d/2$ and $d/5$ dimensions, respectively. As can be seen in Fig. 11, while performance is relatively similar for the different approaches, we notice that increasing the number of $\mathbf{R}$ matrices does not necessarily increase performance, and can instead lead to performance degradation. This is not surprising as we are considering the clustering setting, where we lack strong supervised guidance, and increasing complexity and model flexibility could potentially result in degenerate solutions. We thus recommend learning a single shared $\mathbf{R}$.

## C.7 DISCUSSION ON EXTREME SCENARIOS

To further evaluate the robustness of our proposed approach, we also consider more extreme missing-rate scenarios. Figure 13 provides performance curves vs missing-rate for an extended range of [0,0.9] for the Handwritten and Scene15 datasets. Results demonstrate that the proposed approach consistently outperforms the baselines even in the two extreme cases (missing rate of 90% and of 0% (complete)), further supporting the need to break the independence assumption.

## D LIMITATION AND FUTURE WORK

While our method demonstrates effectiveness on incomplete multi-view clustering tasks by relaxing the independence assumption and adaptively modeling cross-view correlations, it has practical limitations when it comes to the computational cost of scaling to a substantially larger number of views (*e.g.*, $V > 20$). While most current settings leverage only a small number of views, this

could potentially become a limitation in the future. In Appendix A.7, we briefly outline how our algorithm can be implemented more efficiently to also address large $V$ scenarios. Besides, in the future, we would like to further investigate the application of our proposed mechanism in the context of imputation-based IMVC methods to further enhance performance. Furthermore, although we follow prior approaches and assume Gaussian distributions in our approach, there remains potential in exploring more expressive or non-Gaussian distributions for more complex data.

## E  BROADER IMPACT

ACOVA is a general approach to aggregate view-specific posteriors and has thus significant potential to be integrated into not only other IMVC approaches but could also alleviate the need for expensive grid-searches and the restrictions of fixed correlation values in other (also supervised) multi-view or multi-modal settings.

## F  LLMS USAGE

In the course of this research and in preparing the manuscript, we utilized Large Language Models (LLMs) in a limited capacity. Specifically, during the manuscript preparation phase, an LLM was used to assist in refining the wording and improving the clarity of the English prose. Its role in this capacity was strictly limited to enhancing sentence structure, grammar, and the overall flow of the text. Beyond this, LLMs were not involved in the research design, data collection, evaluation, or the generation of core scientific ideas. All substantive content, methodologies, and conclusions are entirely the original work of the authors.

