# OpenReview forum: "Breaking Independence: Learning Correlated Views for Variational Incomplete Multi-View Clustering"
_ICLR.cc/2026/Conference — Submitted to ICLR 2026_

### Official Review · Reviewer_DFgg · 2025-10-27

**Soundness:** 3
**Presentation:** 3
**Contribution:** 2
**Rating:** 6
**Confidence:** 4

**Summary:**

This paper challenges the independence assumption in multi-view learning, arguing that in real-world scenarios, different views are often statistically correlated rather than independent. The authors propose a new framework that explicitly models inter-view correlations, achieving better results on various benchmarks.

**Strengths:**

1.The proposed framework introduces a clear mathematical mechanism to capture inter-view dependencies, instead of assuming factorized likelihoods.

2.The derivations are formal, and empirical results confirm that dependency modeling improves both accuracy and robustness.

**Weaknesses:**

1.Experiments only consider moderate missing ratios (30–70%), not extreme cases like 90%. When co-occurrence among views becomes sparse, correlation estimation becomes unreliable.

2.Although the paper claims to “jointly learn correlation structures,” the learned parameters correspond to a global covariance, capturing only linear, global dependencies.

3.The text claims experiments are conducted on four datasets, while tables clearly list five. This inconsistency undermines experimental clarity and should be corrected.

**Questions:**

1.Can your method handle extreme missing-view ratios (e.g., 90%)? Could you provide performance curves versus missing ratios?

2.The learned correlation structure is always shared globally. Can it happen that view dependencies behave differently for different samples? Is a single correlation matrix no longer a good fit?

---

> ### Author Response · Authors · 2025-11-20
> **Individual Response to Reviewer DFgg**
>
> Thank you for the thorough review and constructive feedback. Below, we address your mentioned weaknesses (W) and questions (Q), referring to modifications made in the manuscript (highlighted in blue).
>
> ## W1 and Q1: Performance under extreme missing ratios and Additional Performance Curves
>
> Indeed, ACOVA is not restricted to low-missingness settings and fully functions also under extreme missing ratios. We have followed your suggestion to test this empirically and have included performance curves vs. missing rates for the Handwritten and Scene-15 datasets in Appendix C.7 (Figure 13). In this figure, we see that the proposed method maintains its substantial advantages over competing methods also for more extreme cases such as 90\% missing rate as well as for the complete scenario (0\%). This further supports the importance of breaking the independence assumption.
>
> ## W2 and Q2: Global linear and sample-specific dependencies
>
> Thank you for this comment and the opportunity to clarify our claim.
>
> Our goal is to learn cross-view dependency structure in the variational posterior, not data-dependent nonlinear correlations. Thus, "jointly learn correlation structures" refers specifically to the learned inter-view covariance of the variational distributions, which captures how representations from different views relate to one another. To avoid misunderstanding, we revised the text to state that we are "jointly learning a global, latent-space correlation matrix" when listing our contributions in the Introduction.
>
> Note, while R captures linear dependencies, this is done in latent space, with each view's encoder (neural network with ReLU) producing nonlinear latent features. Consequently, the overall model still expresses nonlinear, structured cross-view interactions. This departs from the common assumption of view independence and is supported by the consistent improvements in Table 1.
>
> While different samples may indeed exhibit different cross-view relationships, our focus is intentionally on learning the shared global structure, analogous to how VAEs employ a global prior. This promotes stable training and identifiability (see newly added Appendix A.3 for proof of identifiability). Moreover, the non-linear encoders provide some flexibility for capturing sample-specific variability within that shared structure. However, more explicitly modeling instance-specific or non-linear correlations could be an interesting avenue of future work.
>
> ## W3: Typos Correction.
> Thank you for pointing out the typos. We have fixed them and had a new pass over the manuscript to correct other typos.
>
> ---
> We sincerely thank you again for your review and efforts! Your comments have greatly strengthened our work, and we hope our revisions address your concerns effectively.

---

### Official Review · Reviewer_h14L · 2025-10-31

**Soundness:** 3
**Presentation:** 2
**Contribution:** 3
**Rating:** 6
**Confidence:** 5

**Summary:**

This paper addresses the limitations of existing imputation-free variational methods for incomplete multi-view clustering (IMVC), which typically rely on the assumption of conditional independence across views.

The authors propose ACOVA (Adaptive Correlation-aware Variational Aggregation), a novel variational framework that explicitly models and learns inter-view correlations by leveraging the covariance of estimation errors between view-specific posteriors. The correlation matrix is parameterized through a normalized Cholesky decomposition, ensuring positive definiteness and enabling joint end-to-end optimization with model parameters.

Comprehensive experiments on several standard IMVC benchmarks (Scene15, Caltech5V, Handwritten, Fashion-MV, NoisyMNIST) show that ACOVA consistently outperforms previous state-of-the-art methods under various missing-view settings. Ablation and visualization analyses further confirm the benefit of modeling adaptive correlations for robust and discriminative latent representation learning.

**Strengths:**

- The paper identifies a fundamental issue in variational IMVC — the conditional independence assumption among view posteriors. Proposing to “break independence” by modeling cross-view correlation of estimation errors is both theoretically motivated and empirically useful.
- Built upon a linear–Gaussian variational framework, the proposed ACOVA explicitly models structured posterior covariance by decomposing the estimation error covariance as Σ=DRD.  The normalized Cholesky parameterization of R captures the correlations among view-specific variances and enables joint optimization, providing a principled generalization from DVIMC (independent) → CoDE (fixed scalar correlation) → ACOVA (adaptive correlation).
- Experiments on five IMVC benchmarks demonstrate consistent gains across missing-view rates (10–70%), confirming the method’s robustness. Qualitative visualizations — including t-SNE plots and learned correlation matrices — clearly illustrate that the model captures meaningful inter-view dependencies.
- Overall, the paper — together with its appendices — forms a coherent and complete study, covering theoretical justification, empirical validation, and in-depth analysis.

**Weaknesses:**

Motivation
1. The work is focused narrowly on incomplete multi-view clustering (IMVC), a relatively specific subproblem. It would strengthen the contribution to discuss whether the proposed adaptive correlation learning principle generalizes to broader multi-modal or self-supervised representation learning tasks.
2. The paper defines the “error of estimation” as the bridge to cross-view correlation modeling, but this definition appears somewhat _indirect_. The authors should provide a higher-level motivation early in the introduction — e.g., why modeling estimation errors is the right abstraction for inter-view dependence, rather than just citing Winkler (1981) and Mancisidor et al. (2025).

Method

3. The adaptive correlation learning is elegant, but the derivation (Eq. 9–11) and the optimization of R (Eq. 16) could use _clearer intuition_. It’s not entirely clear how learning R avoids degeneracy when views are highly incomplete or uncorrelated.
4. The paper provides only a Frobenius norm bound for R, which is purely structural and does not guarantee _stability_ or _identifiability_ of learned correlations, especially under high missing ratios.
5. In Eq. 6, since both the diagonal scaling $D$ and the correlation matrix $R$ are learned jointly, and $R$ itself is derived from a normalized Cholesky factor $L$, how do you ensure parameter identifiability? In other words, could different $L$ (or scaling of $D$ ) lead to equivalent $\Sigma=D R D$ and thus yield degenerate solutions?

Experiments

6. The authors mention averaging over five runs, but this should be stated clearly in the main text rather than only in the appendix.
7. I appreciate that you provide a complexity analysis in Appendix A.4 and a time-cost comparison in Appendix C.3. However, the theoretical complexity $𝑂(𝑁𝐷𝑉^3)$ scales cubically with the number of views $𝑉$, primarily due to per-sample matrix inversion. Have you analyzed how the actual runtime or memory cost changes with increasing $𝑉$?
8. Although the proposed framework ensures $R \succ 0$ via the normalized Cholesky parameterization, matrix inversion of $R$ (and thus $\Sigma=D R D$ ) is required per sample and per latent dimension. This could introduce numerical instability when $R$ becomes ill-conditioned during training.
9. All datasets used are standard, small to medium-scale benchmarks (Scene15, Caltech5V, Handwritten, Fashion-MV, NoisyMNIST). There are no experiments on high-dimensional or large multi-view datasets to demonstrate scalability or generalization capability.

If the authors can adequately address the above concerns, I would be inclined to raise my score.

**Questions:**

Please refer to weaknesses section, thanks.

---

> ### Author Response · Authors · 2025-11-20
> **(1/3) Individual Response to Reviewer h14L**
>
> Thank you for the very useful and thorough suggestions and comments, which have helped us further strengthen our submission. Below, we address your comments one-by-one and link to the relevant changes in the manuscript. Changes in the manuscript are highlighted in blue.
>
> ## W1  Generalization beyond IMVC
>
>
> Thank you for the comment. While we briefly discussed the more general applicability of ACOVA beyond IMVC in our Broader Impact section (Appendix E), we have in the rebuttal taken several steps towards demonstrating this.
>
> In the revised manuscript, we further demonstrate that our approach is modality agnostic and can also be leveraged to cluster multi-modal data. In particular, we have added an additional multi-modal image-text dataset (CUB Image-Caption) in Table 1.  Results on this dataset provide further evidence for the benefit of modeling inter-view dependencies.
> The results are included here for convenience.
>
> | Missing Rate | **10%** | | | | **30%** | | | | **50%** | | | | **70%** | | | |
> |------------|---------|---------|---------|---------|---------|---------|---------|---------|---------|---------|---------|---------|---------|---------|---------|---------|
> |  **Metrics** | **ACC**↑ | **NMI**↑ | **ARI**↑ | **PUR**↑ | **ACC**↑ | **NMI**↑ | **ARI**↑ | **PUR**↑ | **ACC**↑ | **NMI**↑ | **ARI**↑ | **PUR**↑ | **ACC**↑ | **NMI**↑ | **ARI**↑ | **PUR**↑ |
> | BSV | 0.6723 | 0.6604 | 0.4818 | 0.6823 | 0.6080 | 0.5858 | 0.3376 | 0.6163 | 0.5270 | 0.5241 | 0.2295 | 0.5387 | 0.4493 | 0.4474 | 0.1266 | 0.4630 |
> | CONCAT | 0.6890 | 0.6685 | 0.4979 | 0.6947 | 0.5837 | 0.5790 | 0.3322 | 0.5987 | 0.5187 | 0.5107 | 0.2068 | 0.5323 | 0.4663 | 0.4658 | 0.1391 | 0.4770 |
> | DSIMVC | 0.5227 | 0.5822 | 0.2867 | 0.5833 | 0.5163 | 0.5781 | 0.3008 | 0.5797 | 0.4040 | 0.4430 | 0.1196 | 0.4587 | 0.2653 | 0.2337 | 0.0571 | 0.2867 |
> | DCP |  0.6687 | 0.6764 | *0.6356* | 0.5396 | 0.5560 | 0.5603 | *0.5767* | 0.3731 | 0.5497 | 0.5340 | *0.5100* | 0.2823 | 0.4830 | 0.4810 | 0.4547 | 0.1933 |
> | CSPAN  | 0.6380 | 0.5744 | 0.4280 | 0.6427 | 0.6420 | 0.5663 | 0.4236 | 0.6420 | 0.5810 | 0.5419 | 0.3824 | 0.5860 | 0.5410 | 0.4965 | 0.3275 | 0.5440 |
> | DVIMC | 0.6573 | 0.6585 | 0.5246 | 0.6807 | 0.6703 | 0.6502 | 0.5131 | 0.6807 | 0.6057 | 0.6026 | 0.4538 | 0.6160 | 0.4710 | 0.4694 | 0.2795 | 0.4803 |
> | MVP | 0.6497 | 0.6317 | 0.5051 | 0.6667 | *0.6840* | 0.6545 | 0.5256 | 0.6903 | 0.6163 | 0.5750 | 0.4401 | 0.6287 | *0.6440* | *0.6036* | *0.4655* | *0.6510* |
> | PMIMC | *0.7183* | *0.7131* | 0.5795 | *0.7383* | 0.6820 | *0.6847* | 0.5360 | *0.6977* | *0.6517* | *0.6198* | 0.4846 | *0.6590* | 0.5907 | 0.5480 | 0.4034 | 0.6060 |
> | ACOVA | **0.7970** | **0.7695** | **0.6581** | **0.7970** | **0.7563** | **0.7335** | **0.6215** | **0.7590** | **0.6823** | **0.6568** | **0.5267** | **0.6917** | **0.6554** | **0.6136** | **0.4730** | **0.6584** |
>
>
> We have further added results in Appendix C.7, demonstrating that the proposed approach also generalizes to the complete multi-view clustering setting (0\% missing rate).
>
> ## W2 Additional motivation in Introduction
>
> Thank you for pointing this out, we have revised the Introduction to provide additional high-level motivation on why we leverage the error of estimation (highlighted in blue in the revised manuscript).

---

> ### Author Response · Authors · 2025-11-20
> **(2/3) Individual Response to Reviewer h14L**
>
> ## W3 Avoiding Degeneracy of R
> Thank you for making us aware that this was not explicitly stated. We have now added more intuition to Eq. 9-11 and in particular the role of the binary mask $\mathbf{M}$, which indicates view availability. This mask directly influences the gradients that flow to $\mathbf{R}$, zeroing out all contributions from missing views. Further, by parameterizing $\mathbf{R}$ through the Cholesky decomposition, we constrain it to the space of valid correlation matrices, thus avoiding degenerate solutions.
>
> In the following, we further elaborate on stability and identifiability, providing both theoretic results as well as additional empirical results demonstrating that the proposed approach indeed avoids degeneracy even when views are highly incomplete or uncorrelated.
>
>
> ## W4 and W5: Identifiability and stability
>
>
> Thank you for raising this interesting point. In the updated version, we have added a new theorem and proof (see Appendix A.3) that demonstrates identifiability of our proposed covariance parametrization, by leveraging the fact that $\mathbf{R}$  is a correlation matrix and that the Cholesky factor is unique.
>
>
> In addition, we have conducted empirical validation, both demonstrating convergence stability for high missing ratios (90\%) in Figure 8 and evaluating identifiability in Figure 12 (Appendix C.5). For the latter, we conduct five runs with different seeds of the Handwritten dataset and show that $\mathbf{R}$ exhibits low variance across independent runs, consistently converging to the same solution. The $\mathbf{R}$ matrix for 90\% and 10\% missing rates are included below.
>
>
> **R Matrix (Mean ± Std) for 90\% Missing Rate**
>
> |           | View 1           | View 2           | View 3           | View 4           | View 5           | View 6           |
> |-----------|------------------|------------------|------------------|------------------|------------------|------------------|
> | **View 1**| 1.000 ± 0.000    | 0.282 ± 0.001    | 0.296 ± 0.001    | 0.274 ± 0.006    | 0.280 ± 0.001    | 0.206 ± 0.003    |
> | **View 2**| 0.282 ± 0.001    | 1.000 ± 0.000    | 0.276 ± 0.002    | 0.274 ± 0.004    | 0.281 ± 0.000    | 0.187 ± 0.002    |
> | **View 3**| 0.296 ± 0.001    | 0.276 ± 0.002    | 1.000 ± 0.000    | 0.288 ± 0.010    | 0.260 ± 0.001    | 0.210 ± 0.003    |
> | **View 4**| 0.274 ± 0.006    | 0.274 ± 0.004    | 0.288 ± 0.010    | 1.000 ± 0.000    | 0.270 ± 0.003    | 0.209 ± 0.003    |
> | **View 5**| 0.280 ± 0.001    | 0.281 ± 0.000    | 0.260 ± 0.001    | 0.270 ± 0.003    | 1.000 ± 0.000    | 0.190 ± 0.005    |
> | **View 6**| 0.206 ± 0.003    | 0.187 ± 0.002    | 0.210 ± 0.003    | 0.209 ± 0.003    | 0.190 ± 0.005    | 1.000 ± 0.000    |
>
>
> **R Matrix (Mean ± Std) for 10\% Missing Rate**
> |           | View 1           | View 2           | View 3           | View 4           | View 5           | View 6           |
> |-----------|------------------|------------------|------------------|------------------|------------------|------------------|
> | **View 1**| 1.000 ± 0.000    | 0.641 ± 0.011    | 0.675 ± 0.007    | 0.613 ± 0.018    | 0.630 ± 0.008    | 0.270 ± 0.016    |
> | **View 2**| 0.641 ± 0.011    | 1.000 ± 0.000    | 0.691 ± 0.005    | 0.598 ± 0.014    | 0.705 ± 0.005    | 0.241 ± 0.021    |
> | **View 3**| 0.675 ± 0.007    | 0.691 ± 0.005    | 1.000 ± 0.000    | 0.617 ± 0.015    | 0.687 ± 0.005    | 0.299 ± 0.031    |
> | **View 4**| 0.613 ± 0.018    | 0.598 ± 0.014    | 0.617 ± 0.015    | 1.000 ± 0.000    | 0.609 ± 0.005    | 0.333 ± 0.015    |
> | **View 5**| 0.630 ± 0.008    | 0.705 ± 0.005    | 0.687 ± 0.005    | 0.609 ± 0.005    | 1.000 ± 0.000    | 0.288 ± 0.024    |
> | **View 6**| 0.270 ± 0.016    | 0.241 ± 0.021    | 0.299 ± 0.031    | 0.333 ± 0.015    | 0.288 ± 0.024    | 1.000 ± 0.000    |
>
>
> ## W6: Statement of five-run averaging
>
> Thank you for pointing out that this was not stated clearly enough. While we mentioned this briefly in Section 5.1 (Incomplete multi-view data processing), we have highlighted this further in the revised version by including it also in the captions of Table 1 and 2.

---

> ### Author Response · Authors · 2025-11-20
> **(3/3) Individual Response to Reviewer h14L**
>
> ## W7: Runtime changes with increasing V
>
> In the revised version, we have now extended the time-cost analysis in Appendix C.3 to explicitly measure the impact of increasing the number of views ($V$). For this we take the Caltech5V dataset that contains 5 views and measure the time per epoch when including different numbers of views. The results are also provided here for your convenience below. While $V$ is a bottleneck for very large number of views, in most practical settings where views are few, the cubic cost does not pose a significant computational burden. Note, while the number of views is typically small, more efficient implementations could be considered that avoid explicitly computing the inverse (see next response).
>
>
> | Number of Views (V) | 2V           | 3V           | 4V           | 5V           |
> |---------------------|--------------|--------------|--------------|--------------|
> | Runtime (seconds)   | 0.18 ± 0.03  | 0.22 ± 0.03  | 0.26 ± 0.04  | 0.27 ± 0.04  |
>
> ## W8: Matrix inversion of R if ill-conditioned
>
> Thank you for raising this concern. In our experiments, we have not encountered stability issues and have therefore not focused too much on this aspect in the submission. However, we understand that the matrix inversion can raise concerns and have therefore added an additional discussion in Appendix A.7 that outlines a more stable and computationally more efficient alternative implementation of our algorithm that avoids the explicit computation of the inverse.
>
> ## W9: More complex datasets
> In the revised version we have now added a sixth dataset, namely the CUB Image-Caption dataset. This dataset tests our approach on a more complex multi-modal dataset (images+text), as also requested in W1. Results are provided in the updated Table 1 and our reply to W1.
>
>
> ---
> Once again, we are sincerely grateful for your efforts and your constructive and detailed feedback, which has helped us strengthen the manuscript. We hope our revisions now fully meet with your approval.

---

> > ### Comment · Reviewer_h14L · 2025-11-21
> > **Response to the authors**
> >
> > Good to go. I read through the rebuttal thoroughly, and most of what I was worried about is covered. I’m raising my score to 8, but make sure all these edits are carefully included in the next draft.

---

> > > ### Author Response · Authors · 2025-11-21
> > >
> > > Thank you again for your useful suggestions and for acknowledging the rebuttal and increasing your score.

---

### Official Review · Reviewer_AN2K · 2025-11-01

**Soundness:** 3
**Presentation:** 2
**Contribution:** 3
**Rating:** 4
**Confidence:** 5

**Summary:**

This paper targets the problem of Incomplete Multi-View Clustering (IMVC). It identifies a key limitation in recent variational IMVC approaches: their reliance on aggregators that assume conditional independence between views, which is a potentially over-restrictive assumption in real-world multi-view scenarios. To alleviate this limitation, the paper proposes ACOVA, a variational framework that relaxes this assumption by explicitly modeling cross-view correlations to achieve a more robust representation aggregation method. The core mechanism involves learning a correlation matrix, which is parameterized to ensure positive definiteness and trained end-to-end within a unified variational objective. Experimental results on several datasets demonstrate that ACOVA achieves superior performance, particularly surpassing methods that rely on an independence assumption.

**Strengths:**

The paper identifies a practical limitation in prior variational IMVC work: the conditional independence assumption inherent in the aggregation method, especially in the Product-of-Experts (PoE) approach, which might be over-restrictive in many multi-view scenarios, thereby limiting such IMVC method performance. The proposed solution of parameterizing a learnable correlation matrix via Cholesky decomposition (to ensure positive definiteness) is an elegant and technically sound approach to address this limitation in an unsupervised manner. Additionally, the experiments reveal the effectiveness of explicitly modeling and incorporating inter-view correlations.

**Weaknesses:**

1. Several key notations in the methodology (Sec 4.1) are ambiguous or insufficiently defined, which significantly hinders the comprehension of the proposed method. For example, the definition of $\mu$ as a $\mathbb{R}^{VD\times1}$ column vector is confusingly represented as $[\mu^1, \mu^2, ..., \mu^D]^{\mathrm{T}}$. Furthermore, the precise structure of the design matrix $\mathbb{1}$ in Eq.5 is not clearly specified; it appears to be a block matrix where the $d$-th $V$-row block contains ones in the $d$-th column and zeros elsewhere, but this requires explicit definition. Additionally, terms like $\mathbf{A}_\mathbf{M}$, $\mathbf{M}$ in Eq.9-11 are used in the main text without sufficient introduction.
2. While the current experiments demonstrate the method's effectiveness to some extent, the chosen multi-view datasets (e.g., Handwritten, Fashion) appear relatively simple, with potentially obvious inter-view correlations. To further validate the robustness and generalizability of the proposed correlation modeling, the method should be evaluated on more complex, real-world multi-view datasets where inter-view relationships may be more subtle or heterogeneous.

**Questions:**

1.Modeling view correlation seems fundamentally beneficial. Theoretically, should this approach also be expected to outperform independence-assuming methods in the complete multi-view setting?

2.The current datasets seem to have relatively simple view structures. How is the method expected to perform on more complex heterogeneous views (e.g., involving different modalities) where the inter-view correlations might be more intricate?

---

> ### Author Response · Authors · 2025-11-20
> **Individual Response to Reviewer AN2K**
>
> Thank you for your constructive comments and thorough review. Below, we address your mentioned weaknesses (W) and questions (Q), referring to modifications made in the manuscript (highlighted in blue).
>
> ## W1 Clarification of notations and definitions.
>
> We appreciate your careful comments and observations regarding notation clarity. We acknowledge that some notations in Section 4.1 would benefit from better definitions and detailed explanation. In the revised manuscript, we have revised the definition of $\mu$,$\mu^D$, and $\mu_V^D$, clarifying their dimensions. We also provide more details on $ \mathbb{1} $ , $ \mathbf{A}\_\mathbf{M} $ , $ \mathbf{B}\_\mathbf{M} $ and $\mathbf{M}$ in the updated Section 4.1.
>
> ## W2 and Q2: Experiments on more complex, real-world multi-view dataset.
>
> Thank you for this suggestion. We have now added another heterogeneous dataset called CUB Image-Caption, which includes complex images of birds from Caltech-UCSD Birds and corresponding text captions describing the birds’ appearance characteristics collected by Amazon Mechanical Turk. We follow the setup in [1,2]. Results are included in the updated Table 1, with the proposed ACOVA outperforming the baselines. We show the table below for the your convenience.
>
>
> | Missing Rate | **10%** | | | | **30%** | | | | **50%** | | | | **70%** | | | |
> |------------|---------|---------|---------|---------|---------|---------|---------|---------|---------|---------|---------|---------|---------|---------|---------|---------|
> |  **Metrics** | **ACC**↑ | **NMI**↑ | **ARI**↑ | **PUR**↑ | **ACC**↑ | **NMI**↑ | **ARI**↑ | **PUR**↑ | **ACC**↑ | **NMI**↑ | **ARI**↑ | **PUR**↑ | **ACC**↑ | **NMI**↑ | **ARI**↑ | **PUR**↑ |
> | BSV | 0.6723 | 0.6604 | 0.4818 | 0.6823 | 0.6080 | 0.5858 | 0.3376 | 0.6163 | 0.5270 | 0.5241 | 0.2295 | 0.5387 | 0.4493 | 0.4474 | 0.1266 | 0.4630 |
> | CONCAT | 0.6890 | 0.6685 | 0.4979 | 0.6947 | 0.5837 | 0.5790 | 0.3322 | 0.5987 | 0.5187 | 0.5107 | 0.2068 | 0.5323 | 0.4663 | 0.4658 | 0.1391 | 0.4770 |
> | DSIMVC | 0.5227 | 0.5822 | 0.2867 | 0.5833 | 0.5163 | 0.5781 | 0.3008 | 0.5797 | 0.4040 | 0.4430 | 0.1196 | 0.4587 | 0.2653 | 0.2337 | 0.0571 | 0.2867 |
> | DCP |  0.6687 | 0.6764 | *0.6356* | 0.5396 | 0.5560 | 0.5603 | *0.5767* | 0.3731 | 0.5497 | 0.5340 | *0.5100* | 0.2823 | 0.4830 | 0.4810 | 0.4547 | 0.1933 |
> | CSPAN  | 0.6380 | 0.5744 | 0.4280 | 0.6427 | 0.6420 | 0.5663 | 0.4236 | 0.6420 | 0.5810 | 0.5419 | 0.3824 | 0.5860 | 0.5410 | 0.4965 | 0.3275 | 0.5440 |
> | DVIMC | 0.6573 | 0.6585 | 0.5246 | 0.6807 | 0.6703 | 0.6502 | 0.5131 | 0.6807 | 0.6057 | 0.6026 | 0.4538 | 0.6160 | 0.4710 | 0.4694 | 0.2795 | 0.4803 |
> | MVP | 0.6497 | 0.6317 | 0.5051 | 0.6667 | *0.6840* | 0.6545 | 0.5256 | 0.6903 | 0.6163 | 0.5750 | 0.4401 | 0.6287 | *0.6440* | *0.6036* | *0.4655* | *0.6510* |
> | PMIMC | *0.7183* | *0.7131* | 0.5795 | *0.7383* | 0.6820 | *0.6847* | 0.5360 | *0.6977* | *0.6517* | *0.6198* | 0.4846 | *0.6590* | 0.5907 | 0.5480 | 0.4034 | 0.6060 |
> | ACOVA | **0.7970** | **0.7695** | **0.6581** | **0.7970** | **0.7563** | **0.7335** | **0.6215** | **0.7590** | **0.6823** | **0.6568** | **0.5267** | **0.6917** | **0.6554** | **0.6136** | **0.4730** | **0.6584** |
>
>
> [1] Shi Y, Paige B, Torr P. Variational mixture-of-experts autoencoders for multi-modal deep generative models. Advances in neural information processing systems, 2019, 32.
>
> [2] Zhang C, Han Z, Cui Y, Fu H, Zhou J T, Hu Q. CPM-Nets: Cross Partial Multi-View Networks. Advances in neural information processing systems, 2019, 32.
>
> ## Q1: Comparison with baseline in the complete multi-view setting
>
> This is a very good point and modeling the correlation should also be beneficial for the complete multi-view setting (0\% missingness). We have experimentally validated this on the Handwritten and Scene-15 datasets during the rebuttal by considering a broader range of missingness settings ranging from 0\% to 90\%. Results are provided in Figure 13 of Appendix C.7 and show that also under the complete setting (0\%), ACOVA achieves superior performance on both datasets, indicating the importance of exploiting cross-view relationships.
>
> In summary, modeling view correlation outperforms independence-assuming methods in both complete and incomplete settings.
>
>
>
> ---
> We hope this resolves your concerns, and we sincerely thank you once again for your careful review and thoughtful comments, which have been instrumental in strengthening our work. We hope the revised document fully addresses your concerns.

---

### Official Review · Reviewer_XMzh · 2025-11-03

**Soundness:** 2
**Presentation:** 3
**Contribution:** 2
**Rating:** 2
**Confidence:** 5

**Summary:**

The paper presents an IMVC model by introducing a variational framework that explicitly models and learns cross-view correlations, thereby addressing a significant limitation of existing imputation-free methods based on variational inference.  Overall, with more extensive experimental validation, sensitivity analysis, discussion of limitations, and visualizations, this paper has the potential to make a substantial impact in the field.

**Strengths:**

1. The introduction of a learnable cross-view correlation structure is a significant contribution to the field of IMVC.
2. The authors designed an adaptive cross-view correlation learning mechanism to jointly learn the correlation matrix and model parameters, leading to improved performance

**Weaknesses:**

1. It is suggested to perform more comparison experiments with state-of-the-art methods on a wider range of datasets and evaluation metrics. This would help to solidify the claims about the robustness and efficiency of the proposed approach.
2. The advancement of this paper and the derivation of the intermediate process are expected to receive more mathematical support, such as Eq. (1)-(3).

**Questions:**

1. What is the complexity of the proposed method? How does it compare with other methods, especially on the large-scale datasets?
2. How does it perform on the data with complex noise? How to ensure the accuracy of the learned cross-view correlation structure?
3. How many hyperparameters are there in the proposed model, and how can they be set in a new IMVC task?
4. How does the computational complexity scale with the number of views or the size of the dataset? Are there any scenarios where the proposed method might not perform well?

---

> ### Author Response · Authors · 2025-11-20
> **(1/3) Individual Response to Reviewer XMzh**
>
> Thank you for your comments and for recognizing that we address a significant limitation in current approaches that has the potential to make a substantial impact on the field. Below, we address your mentioned weaknesses (W) and questions (Q), referring to modifications made in the manuscript (highlighted in blue). Finally, we provide a brief summary.
>
> ## W1: More comparison experiments validation
>
> In the revised document, we have extended the experimental comparisons to include a sixth dataset, CUB Image-Caption. Also on this dataset our proposed approach outperforms the eight state-of-the-art IMVC baselines, further demonstrating the benefit of the proposed method. Below, we include the Table for the reviewers' convenience. The table shows the clustering performance of the CUB Image-Caption dataset. Note: Best results and second best are highlighted in **bold** and *italic*, respectively. All results are averaged over five runs.
>
> | Missing Rate | **10%** | | | | **30%** | | | | **50%** | | | | **70%** | | | |
> |------------|---------|---------|---------|---------|---------|---------|---------|---------|---------|---------|---------|---------|---------|---------|---------|---------|
> |  **Metrics** | **ACC**↑ | **NMI**↑ | **ARI**↑ | **PUR**↑ | **ACC**↑ | **NMI**↑ | **ARI**↑ | **PUR**↑ | **ACC**↑ | **NMI**↑ | **ARI**↑ | **PUR**↑ | **ACC**↑ | **NMI**↑ | **ARI**↑ | **PUR**↑ |
> | BSV | 0.6723 | 0.6604 | 0.4818 | 0.6823 | 0.6080 | 0.5858 | 0.3376 | 0.6163 | 0.5270 | 0.5241 | 0.2295 | 0.5387 | 0.4493 | 0.4474 | 0.1266 | 0.4630 |
> | CONCAT | 0.6890 | 0.6685 | 0.4979 | 0.6947 | 0.5837 | 0.5790 | 0.3322 | 0.5987 | 0.5187 | 0.5107 | 0.2068 | 0.5323 | 0.4663 | 0.4658 | 0.1391 | 0.4770 |
> | DSIMVC | 0.5227 | 0.5822 | 0.2867 | 0.5833 | 0.5163 | 0.5781 | 0.3008 | 0.5797 | 0.4040 | 0.4430 | 0.1196 | 0.4587 | 0.2653 | 0.2337 | 0.0571 | 0.2867 |
> | DCP |  0.6687 | 0.6764 | *0.6356* | 0.5396 | 0.5560 | 0.5603 | *0.5767* | 0.3731 | 0.5497 | 0.5340 | *0.5100* | 0.2823 | 0.4830 | 0.4810 | 0.4547 | 0.1933 |
> | CSPAN  | 0.6380 | 0.5744 | 0.4280 | 0.6427 | 0.6420 | 0.5663 | 0.4236 | 0.6420 | 0.5810 | 0.5419 | 0.3824 | 0.5860 | 0.5410 | 0.4965 | 0.3275 | 0.5440 |
> | DVIMC | 0.6573 | 0.6585 | 0.5246 | 0.6807 | 0.6703 | 0.6502 | 0.5131 | 0.6807 | 0.6057 | 0.6026 | 0.4538 | 0.6160 | 0.4710 | 0.4694 | 0.2795 | 0.4803 |
> | MVP | 0.6497 | 0.6317 | 0.5051 | 0.6667 | *0.6840* | 0.6545 | 0.5256 | 0.6903 | 0.6163 | 0.5750 | 0.4401 | 0.6287 | *0.6440* | *0.6036* | *0.4655* | *0.6510* |
> | PMIMC | *0.7183* | *0.7131* | 0.5795 | *0.7383* | 0.6820 | *0.6847* | 0.5360 | *0.6977* | *0.6517* | *0.6198* | 0.4846 | *0.6590* | 0.5907 | 0.5480 | 0.4034 | 0.6060 |
> | ACOVA | **0.7970** | **0.7695** | **0.6581** | **0.7970** | **0.7563** | **0.7335** | **0.6215** | **0.7590** | **0.6823** | **0.6568** | **0.5267** | **0.6917** | **0.6554** | **0.6136** | **0.4730** | **0.6584** |
>
> We further have performed a more controlled analysis to provide insights into the behavior of our model as missingness increases and noise is added (see Sec. 5.4). We also provide a summary of this experiment in response to your Q2 below.
>
> Besides, we also extend our methods to more extreme scenarios, including complete (0\%) and extremely high missing ratios (90\%) and show the superiority of our methods compared to other approaches, validating the effectiveness of considering and learning correlations (see Appendix C.7).
>
> Note, for metrics, we follow prior work on IMCV, reporting the union of metrics in prior works [1,2,3] (ACC, NMI, ARI), [4](ACC, NMI, ARI, PUR).
>
>
> [1] Lin Y, Gou Y, Liu Z, Li B, Lv J, Peng X. Completer: Incomplete multi-view clustering via contrastive prediction. In CVPR, 2021, 11174-11183.
>
> [2] Gao X, Pu J. Deep incomplete multi-view learning via cyclic permutation of VAEs. In ICLR, 2025.
>
> [3] Xu J, Li C, Ren Y, Peng L, Mo Y, Shi X, Zhu X. Deep incomplete multi-view clustering via mining cluster complementarity. In AAAI, 2022, 8761-8769.
>
> [4] Xu G, Wen J, Liu C, Hu B, Liu Y, Fei L, Wang W. Deep variational incomplete multi-view clustering: Exploring shared clustering structures. In AAAI, 2024, 16147-16155.
>
> ## W2: More details on the mathematical background for Eqs. (1)–(3)
>
> Thank you for making us aware that Sec. 3, where we consider preliminaries from prior work, could be more accessible for readers less familiar with prior variational approaches to IMVC. To improve this aspect, we have expanded the discussion of mathematical background in Appendix A.1 by summarizing the derivations and key intermediate steps that lead to Eqs. (1)–(3). This addition provides the requested mathematical support without interrupting the flow of the main narrative. We believe this clarification will strengthen the paper and appreciate the reviewer’s recommendation.

---

> ### Author Response · Authors · 2025-11-20
> **(2/3) Individual Response to Reviewer XMzh**
>
> ## Q1 and Q4: Complexity analysis
>
> Thank you for emphasizing the importance of the computational complexity analysis. In the original version, we have analyzed the computational complexity in Appendix A.6 and also provided a time-cost comparison (practical runtime of our and other methods on the Scene15 dataset) in Appendix C.3. Runtime results demonstrate that the proposed method only adds a minimal overhead compared to not considering dependence (DVIMC) and is more efficient than state-of-the-art models MVP and PMIMC.
>
> Below, we: (1) briefly revisit the complexity, particularly focusing on large-scale datasets and (2) provide additional empirical runtime results demonstrating scalability on larger datasets and scalability with regards to increasing views.
>
> **1. Computational complexity analysis, especially on large-scale datasets.**
>     The overall complexity is $\mathcal{O}(NDV^3)$, where $N$ denotes the number of samples, $D$ denotes a common latent dimension, and $V$ denotes the number of views. Therefore, when it comes to scale (large $N$) our proposed approach scales well (linearly) and we have revised Appendix A.6 to highlight this explicitly. Note, since the number of views $V$ is typically small in practice (e.g., $V < 10$ and $V \ll N$), the $V^3$ term does for most practical settings not create a computational bottleneck. But even in the scenario where large number of views are considered, our algorithm can be implemented more efficiently to obtain better scalability with $V$. We have added a discussion of this in Appendix A.7 and updated our discussion of limitations in Appendix D.
>
> **2. Experimental time-cost on a large-scale dataset and increasing views.**
>    Besides the reported small dataset comparison, we additionally test the practical running time on the large-scale dataset NoisyMNIST and show the results in Table 7 in Appendix C.3. For your convenience, we show the results below. It can be seen that due to the linear complexity with respect to $N$, our method is still efficient on large-scale datasets, adding negligible overhead compared to not considering dependence (DVIMC) and being more efficient than MVP and PMIMC.
>
>
> | Method  | BSV    | CONCAT | DSIMVC   | DCP       | CPSPAN   | DVIMC   | MVP     | PMIMC   | ACOVA   |
> |---------|--------|--------|----------|-----------|----------|---------|---------|---------|---------|
> | Time(s) | -      | -      | 3.06×10⁰ | 1.48×10¹  | 2.22×10² | 4.98×10⁰ | 7.61×10¹ | 1.14×10² | 5.59×10⁰ |
> | ACC     | 0.2672 | 0.2805 | 0.6245   | 0.9225    | 0.5486   | 0.8741  | 0.6660  | 0.6044  | 0.9377  |
>
> To demonstrate the impact of $V$, we also added an experiment where we evaluate the time per epoch for different number of views on the Caltech5V dataset. We observe in the Table below (and the updated Appendix C.3) that the overhead of the cubic scaling with respect to $V$ is negligible for most common settings, which tend to have few views.
>
> | Number of Views (V) | 2V           | 3V           | 4V           | 5V           |
> |---------------------|--------------|--------------|--------------|--------------|
> | Runtime (seconds)   | 0.18 ± 0.03  | 0.22 ± 0.03  | 0.26 ± 0.04  | 0.27 ± 0.04  |
>
>
>
> ## Q2: Complex noise
>
> We appreciate your suggestions on exploring our work in noisy scenarios and believe that it will provide interesting insights into the learned dependency structure, greatly strengthening our work.
>
> We therefore conducted controlled simulation experiments in Section 5.4 in the revised document, where we created two views by duplicating original images from the Fashion dataset and simulated view heterogeneity through a mutually exclusive noise injection strategy such that each sample has exactly one corrupted view (50\% in view 1 and 50\% in view 2). We systematically varied pepper noise rates from 0.00 to 1.00 to control heterogeneity and missing rates from 0.00 to 0.70 to simulate incomplete data. Our results demonstrate that the learned correlation matrix $\mathbf{R}$ captures a clear bi-directional decline in cross-view correlations: as noise or missing rates increase, the correlations in $\mathbf{R}$ progressively decrease, validating its ability to adaptively model view relationships under both heterogeneous and incomplete conditions. This experiment underscores the necessity of adaptive correlation learning for realistic multi-view clustering.

---

> ### Author Response · Authors · 2025-11-20
> **(3/3) Individual Response to Reviewer XMzh**
>
> ## Q3: Hyperparameters and sensitivity analysis
>
> In the original submission, we provided an overview over the 10 hyperparameters in Appendix B.4 in Table 6. These include the pretraining and training epochs, batch size, optimizer type, learning-rate scheduler type, network learning rate, prior distribution learning rate, correlation learning rate, the balance parameter $\alpha$, and the latent feature dimension $D$. **Note that only one hyperparameter is newly introduced by our approach** compared to the main baseline (DVIMC; assuming independence). This is the learning rate for the correlation, which was fixed across all datasets in our experiments to $ 1e^{-2} $. On a new task, our suggestion would therefore be to start with this default learning rate. For the remaining parameters, our suggestion would be to take as a starting point one of the configurations presented in Table 6, as they overall seem relatively stable, alternatively perform a parameter sweep which could be informed by the unsupervised loss function.
>
>
> ## Summary
> In the revised manuscript and rebuttal, we have addressed your concerns by substantially expanding the empirical, theoretical, and practical analysis of our method. In particular, we have added an additional dataset, now considering six datasets in total, with the newest one being explicitly multi-modal (see updated Table 1 and discussion above). We have further added an additional analysis, testing the behavior of the approach in a controlled setting under varied conditions (see Sec. 5.4 and discussion above). To strengthen the mathematical foundations, we included more mathematical details for the variational incomplete multi-view clustering background (see A.1) and provided a theoretical perspective on the identifiability of our covariance parameterization (see Appendix A.3). Finally, we have revised the discussion of limitations when it comes to computational complexity, highlighting that the potential challenge is the number of views rather than the number of samples, both theoretically and empirically (see Appendix A.6 and C.3). We have also provided an outline of an improved implementation of our proposed algorithm that reduces the reliance on the number of views further (see Appendix A.7).
>
> ---
> Thank you once again for your review and efforts! We believe that your constructive feedback has significantly strengthened our manuscript and hope that these additions provide the broader validation, deeper analysis, and clearer framing that you highlighted as necessary for the method to make a substantial impact on the field.

---

### Author Response · Authors · 2025-11-27
**General Comment**

We would like to thank the reviewers once again for their insightful comments and for recognizing the novelty and significance of our approach. We have carefully addressed all raised questions and concerns through detailed, point-by-point responses and updates to the manuscript. In particular, we have:
- **Added a sixth, more complex multimodal (image–text) dataset**, further demonstrating the general applicability and robustness of our method (Table 1).
- **Included a new analysis that directly simulates cross-view correlation degradation**, showing that our model captures the expected trends (Sec. 5.4).
- **Provided empirical evidence of stability and a theoretical proof of identifiability** for the proposed correlation learning mechanism (Appendix A.3 and C.5).
- **Expanded experiments to “extreme” IMVC regimes**, including both the complete setting (0% missingness) and high missingness (90%), demonstrating consistent advantages (Appendix C.7).
- **Added a more detailed complexity discussion**, highlighting linear scaling with the number of samples and the method’s suitability for large-scale datasets. We further clarified that the cubic scaling with the number of views is rarely problematic as the number of views is typically significantly lower than the number of samples (Appendix A.6, C.3, D), but also outlined an alternative, more scalable implementation of the proposed algorithm (Appendix A.7).
- **Clarified that the method introduces only one additional hyperparameter** (the learning rate for the correlation structure), which we keep fixed across all datasets.
- **Clarified the intentional focus on modeling global linear structure in latent space**, where non-linearity is introduced through the encoders.
- **Improved clarity and self-containment** by adding background on variational IMVC, refining notation, and strengthening the motivation (Sec. 1, 3, 4, Appendix A.1).

All revisions are integrated into the updated manuscript and highlighted in blue. **We are happy to provide any additional clarifications if further questions arise.**

---

### Author Response · Authors · 2025-12-03
**Summary**

Below is a summary of our contributions, the main concerns raised by the reviewers, and the changes made in response.

---
## **Background**

Incomplete multi-view clustering (IMVC) aims to cluster multi-view data when some views are missing. Variational IMVC methods [1–3] are popular, but they rely on the **unrealistic assumption of independence between views**, limiting their effectiveness.

---
## **Summary of Contributions**


Recognizing this “*fundamental issue*” (Reviewer h14L), we propose the **first variational IMVC framework that explicitly learns cross-view dependencies**, addressing “*a significant limitation of existing methods*” (Reviewer XMzh). We introduce a normalized Cholesky-parameterized covariance structure that models view correlations directly in the latent space. Reviewers describe the approach as “*elegant and technically sound*” (Reviewer AN2K) and “*a significant contribution to the field*” (Reviewer XMzh).

Our extensive empirical study—benchmarks, sensitivity analyses, and visualizations—demonstrates that dependency modeling consistently "*improves both accuracy and robustness*" (Reviewer DFgg). Following reviewer feedback, we further added theoretical results and additional experiments that validate stability, identifiability, and performance under extreme noise or missingness.

Reviewer h14L concluded that our submission “*forms a coherent and complete study, covering theoretical justification, empirical validation, and in-depth analysis*.” Furthermore, all reviewers recognized the importance of modeling view correlation and the strength of our empirical results.

---
## **Main Reviewer Concerns and Responses**

- **Experiments**

  - Reviewers XMzh, AN2K and h14L asked for evaluation on more complex or real-world datasets. We added the CUB Image–Caption dataset, demonstrating ACOVAs robustness on complex multimodal data.

  - Reviewers further sought analyses of noisy conditions (XMzh) and missingness extremes (0\%: AN2K, 90\%: DFgg). We added a controlled noise–missingness simulation and 0\%–90\% missingness experiments, showing consistent and adaptive correlation learning.

  - Reviewers XMzh and h14L raised concerns regarding scalability to large datasets and runtime behavior as the number of views increases. In response, we demonstrated that our approach comes with negligible overhead.

  **All requested experiments were added, and all support our method’s advantages.**

- **Theoretical Analysis**

  - Reviewer h14L was curious about the identifiability, stability, and inversion of R during training. We added a theorem and proof establishing identifiability, empirical convergence studies at 90\% missingness, and outlined a more stable implementation alternative that avoids explicit inversion.
  - Reviewer XMzh requested more mathematical support on Eqs. (1)–(3) and more limitation discussion. In response, we included a new appendix section with clearer derivations to improve accessibility for readers unfamiliar with the preliminaries, and updated our limitation section.

- **Writing Suggestions**

  - Reviewers AN2K, h14L, and DFgg requested clearer notation, improved explanations, and stronger high-level motivation. We revised the relevant sections for clarity, consistency, and readability, and added background material to improve self-containment.

---
[1] Xu G, Wen J, Liu C, Hu B, Liu Y, Fei L, Wang W. Deep variational incomplete multi-view clustering: Exploring shared clustering structures. In AAAI, 2024, 16147-16155.

[2] Gao X, Pu J. Deep incomplete multi-view learning via cyclic permutation of VAEs. In ICLR, 2025.

[3] Yin M, Huang W, Gao J. Shared generative latent representation learning for multi-view clustering. In AAAI, 2020, 6688–6695.

---

### Public Comment · ~Zheming_Xu1 · 2026-03-19
**Comments with a Clarification**

We thank the reviewers and the area chair for their careful evaluation and constructive feedback. We fully respect the final decision.

However, we would like to clarify one point in the meta-review, particularly regarding the characterization of the work as having “limited novelty” or being merely an “incremental advance.” While novelty and impact are ultimately matters of judgment, the written reviews themselves do not explicitly frame the contribution as lacking novelty. Instead, the reviewers acknowledged the conceptual and technical contributions:

* Reviewer XMzh describes the introduction of a learnable cross-view correlation structure as a “significant contribution” with the “potential to make a substantial impact in the field”.
* Reviewer AN2K characterizes the approach as “elegant and technically sound”; addressing a “practical limitation in prior variational IMVC work”.
* Reviewer h14L states that the paper “identifies a fundamental issue in variational IMVC”, “forming a coherent and complete study”.
* Reviewer DFgg highlights the clear mathematical mechanism for capturing inter-view dependencies and confirms empirical improvements from dependency modeling.

The main weaknesses identified across reviews instead concern:
* empirical scope (dataset scale/diversity),
* clarity of notation and exposition,
* scalability and extreme missing ratios,
* and further theoretical analysis (e.g., identifiability, stability).

These are valuable and constructive points, which we worked to address during the rebuttal and will continue to strengthen.

For clarity, ACOVA removes the conditional independence assumption underlying PoE-style variational IMVC, and introduces a structured posterior covariance model with adaptive, positive-definite correlation learning optimized end-to-end. Several reviewers explicitly recognized this conceptual shift.

We share this clarification to ensure that the public summary reflects the substance of the written reviews. We appreciate the reviewers’ insights and will incorporate their suggestions to further improve the work.

---

### Meta-Review · Area_Chair_qSph · 2026-01-06

**Summary:**

This paper proposes ACOVA, a variational framework for incomplete multi-view clustering (IMVC) that relaxes the conditional independence assumption commonly used in prior work by explicitly learning a cross-view correlation structure, where the method parameterizes a correlation matrix via a normalized Cholesky decomposition and optimizes it jointly with the model parameters. Experiments on several benchmarks show improved performance over existing IMVC methods. While Reviewers AN2K, h14L, and DFgg acknowledged the paper’s soundness and potential contribution (scores of 4, 6, and 6, respectively), Reviewer XMzh raised substantial concerns and recommended rejection (score 2). The reviewers collectively identified the critical issues, such as, limited novelty, incremental advance, and insufficient theoretical and empirical validation. Moreover, the core idea of modeling cross-view correlations, while technically sound, was seen as a moderate extension of existing variational IMVC frameworks. Reviewers noted that the formulation builds directly on prior work (e.g., VAE, deep IMVC) and does not introduce a fundamentally new paradigm. After careful consideration of the reviews, the author rebuttal, and the ensuing discussion, the consensus is that the contribution does not meet the high bar for acceptance at ICLR.

**Reviewer Concerns:**

The authors responded diligently to all reviewers, adding experiments on an extra dataset, expanding mathematical details, providing complexity analysis. While the paper tackles a relevant problem and presents a technically coherent approach, it does not meet the high bar for novelty and impact required by ICLR. The contributions are viewed as a solid but incremental improvement over existing variational IMVC methods, and the empirical evaluation remains narrow in scale and diversity. Given the mixed reviews and the presence of a clear rejection recommendation from a confident reviewer, the paper is not deemed ready for publication in its current form.

**Reviewer Scores:**

Reviewer XMzh firmly believes that the paper lacks innovation and contribution. Reviewer h14L has a tendency to give higher scores. The other two reviewers have moderate scores. Taking all the opinions into consideration, AC believes that this work and the rebuttal have made some improvements which addressed specific queries of IMVC, but did not alter the fundamental concerns regarding the paper's incremental nature and limited empirical scope.

---

### Decision · Program_Chairs · 2026-01-26

Reject